



# Suppression of precipitation bias on wind velocity from continuous-wave Doppler lidars

Liqin Jin, Jakob Mann, Nikolas Angelou, and Mikael Sjöholm

Department of Wind and Energy Systems, Technical University of Denmark, Frederiksborgvej 399, 4000 Roskilde, Denmark.

**Correspondence:** Liqin Jin (liqn@dtu.dk)

**Abstract.** In moderate to heavy precipitation, rain droplets may deteriorate Doppler lidars' accuracy for measuring the line-of-sight wind velocity because their projected velocity on the beam direction differs greatly from that of air. Therefore, we propose a method of effectively filtering away the adverse effects of rain on velocity estimation by sampling the Doppler spectra faster than the rain drops' beam transit time. By using a special averaging procedure, we can suppress the rain signal by sampling the spectrum at 3 kHz. On a moderately rainy day with a maximum rain intensity of $4\ \mathrm{mmh}^{-1}$, three ground-based continuous-wave Doppler lidars were used to conduct a field measurement campaign at the Risø campus of the Technical University of Denmark. We demonstrate that the rain bias can effectively be removed by normalizing the noise-flattened Doppler spectra with their peak values before they are averaged down to 50 Hz prior to the determination of the speed. In comparison to the sonic anemometer measurements acquired at the same location, the wind velocity bias at 50 Hz is reduced from up to $-1.58$ $\mathrm{ms}^{-1}$ of the conventional lidar data to $-0.18\ \mathrm{ms}^{-1}$ of the normalized lidar data. This significant reduction of the bias occurs at the minute with the highest amount of rain when the measurement distance of the lidar is 103.9 m with a corresponding probe length being 9.8 m. With the smallest probe length, 1.2 m, the rain-induced bias was only present at the period with the highest rain intensity and was also effectively eliminated with the procedure. Thus the proposed method for reducing the impact of rain on continuous-wave Doppler lidar measurements of air velocity is promising, without requiring much computational effort.

## 1 Introduction

Accurate measurement of wind velocity is crucial for many applications in meteorology and wind energy. For example, precise determination of wind flow plays an important role in reducing loads on critical turbine components and power variations, correcting commonly used models for wind energy assessment, improving the performance of wind turbine controllers, and improving the prediction of the potential wind power extracted from the wind (Davoust et al., 2014; Jena and Rajendran, 2015; Li et al., 2018; Samadianfard et al., 2020; Guo et al., 2022). Besides, wind velocity estimation is also useful for understanding important phenomena, i.e., atmospheric boundary layer flows and wind turbulence (Van Ulden and Holtslag, 1985; Türk and Emeis, 2010; Debnath et al., 2017). Therefore, accurate measurements of wind velocity is required, both in the industrial and academic area.

There are several available instruments capable of measuring wind speed, wind direction and turbulence in wind energy, each with advantages and disadvantages. Compared with the in-situ anemometers, like cup or sonic anemometers installed



on the meteorological masts (met masts), that provide point measurements albeit with an accurate measurement of the wind velocity, Doppler lidars also have the potential to accurately measure the wind flow by precise measurements of the Doppler spectra. For more than a decade, Doppler lidars have been widely used as a more and more reliable, valuable and active optical remote sensing instrument with easier and cost-effective deployment. They have been applied to estimate wind resource both

onshore (Bingöl et al., 2009) and offshore (Sempreviva et al., 2008; Peña et al., 2009; Viselli et al., 2019; Elshafei et al., 2021), both by scanning lidars and profiling lidars (Mann et al., 2017; Menke et al., 2020; Gottschall et al., 2021) with good spatial and temporal resolutions (Henderson et al., 1991; Aoki et al., 2016).

Doppler lidars have the potential of reducing loads on the turbine blade and tower through lidar prevision of the incoming gusts and flow (Bossanyi et al., 2014; Bos et al., 2016), and improving wind turbine control (Mikkelsen et al., 2013; Schlipf

et al., 2015; Zhang and Yang, 2020). Doppler lidars can also be applied to study atmospheric turbulence along the span of a suspension bridge (Cheynet et al., 2016) and study the turbulent wind field in the near-wake region of a tree (Angelou et al., 2022). In order to improve the measurement accuracy by lidars, Wildmann et al. (2020) reduced the volume-averaging effect on the retrieval of the wind flow statistics with ground-based Doppler lidars (see also Sathe and Mann, 2013). Brinkmeyer (2015) suggested the low coherence Doppler lidar approach using a pseudo-random broadband laser source to obtain an effectively

smaller sampling volume. It is self-evident that the precise determination of the wind velocity with Doppler lidars is paramount for many applications in wind energy.

Doppler lidars can be influenced by heavy rainfall because the projected speed of the raindrops on the propagation direction of the lidar beam will be different from the line-of-sight wind velocity. A synergy approach was proposed by (Träumner et al., 2010), which combined radar and vertically scanning lidar measurements to estimate the vertical wind velocity and the raindrop

size distribution during rain episodes. Later, by using a velocity-azimuth display (VAD) scanning technique, wind speed and rainfall speed were simultaneously retrieved in (Wei et al., 2019), by fitting the two-peak spectrum with a two-component Gaussian model. The spectral peak close to $0$ ms$^{-1}$ is the Doppler signal of the vertical wind speed, which can be easily recognized in this scenario. Aoki et al. (2016) and Wei et al. (2021) proposed an iterative deconvolution method to retrieve raindrop size distribution during rain by using a vertically pointing coherent Doppler lidar.

However, for Doppler lidars which are not vertically pointing, the line-of-sight wind velocity is not close to zero and it is difficult to distinguish which part of the signal originated from rain drops or from air-following aerosols. Therefore, the purpose of the present study is to experimentally investigate a method to filter away the precipitation signal from the aerosol signals, in order to reduce the rain-induced bias on the velocity estimation.

A field measurement campaign was carried out at Risø where three coherent cw Doppler lidars (Mikkelsen et al., 2017) were

deployed to point towards a common focus point very close to a mast-mounted sonic anemometer at 31 m height. Each lidar had different elevation angles, focus distances and thus probe lengths. Therefore, it was possible to investigate the influence of these parameters on the performance of the post-processing method. The basic idea is to sample Doppler spectra rapidly, i.e. 3 kHz, which allows us to detect when a raindrop is in the beam and filter out those spectra. Measurements of a sonic anemometer are used as a reference to compare with the estimated line-of-sight wind velocity of the three lidars, before and after filtering away



the rain component in the Doppler spectra. The corresponding rain characteristics is retrieved from a ground-based disdrometer (Tilg et al., 2020) nearby the meteorological mast.

Section 2 introduces the field campaign and elaborately describes the instruments used. In Section 3, the measurement results of the sonic anemometers and the disdrometer are presented. The principle of Doppler spectral processing to retrieve the line-of-sight wind velocity as well as the method we propose to filter away the rain signal are presented in detail in Section 4. Section 5 shows the comparison of 50 Hz and 1-minute wind velocity time series between the lidar and sonic anemometer measurements, with and without filtering away rain signals. The most important findings of our study are summarized in the Conclusion (Section 6).

## 2 Instrumentation

### 2.1 WindScanner lidar system

In order to validate the method to reduce the influence of the precipitation on the estimated wind velocity, we conducted a field experiment at the Risø campus of the Technical University of Denmark (DTU), as shown in Fig. 1. The surrounding terrain is flat and agricultural. The short-range WindScanner lidar system with three cw lidars (Fig. 2) which is developed by DTU Wind and Energy Systems, was used to measure the wind field (Vasiljević et al., 2017; Mikkelsen et al., 2020). The three WindScanners employ a dual-prism beam scanner, enabling them to orient the beam in any direction within $\pm 61°$ of the adjustable center axis (Sjöholm et al., 2014; Mikkelsen et al., 2008). The direction of the line-of-sight of each lidar is steered by two prism motors and a focus motor controls the measurement location along the beam for these lidars. For this campaign, the sampling frequency of spectra is set to be 3 kHz. A central master computer is used to synchronize the short-range wind lidars to scan the same pattern in space simultaneously, however, all the three scanners are focused on one static point in this investigation.

The three ground-based WindScanner lidars were staring at a point 1 m north of a sonic anemometer (USA-1, METEK) which was located 31 m above the ground. The scanner heads were covered with green rain barrels to avoid rain droplets covering the windows of the lidars (Fig. 2). The intention to use three WindScanner lidars is to investigate the influence of different probe lengths and different elevation angles on the performance of the method to filter away rain signals. The full-width-at-half-maximum (FWHM) of the Lorentzian weighting function or the probe length can be approximated as,

$$\text{FWHM} = 2 \cdot z_R = 2 \cdot \frac{\lambda \cdot R^2}{\pi r^2} \tag{1}$$

where $z_R$ is the Rayleigh length, defined as the distance from the focus point to where the cross-sectional area of the laser beam is doubled (Angelou et al., 2012b), $R$ is the distance from the lidar to where the beam is focused, $\lambda$ is the laser wavelength, which is $1.565\ \mu m$, $r$ is the $e^{-2}$ intensity radius of the laser beam at the lidar telescope, which is about 33 mm. A list of the measurement parameters of the WindScanner lidars is summarized in Table 1 and the three-dimensional view, as well as the top view of the configuration of the three lidars is depicted in Fig. 3. WindScanner unit #1 is placed on a slope, therefore it has a relatively bigger elevation angle about $58°$, but has the smallest probe length of 1.2 m, compared with unit #2 and #3.



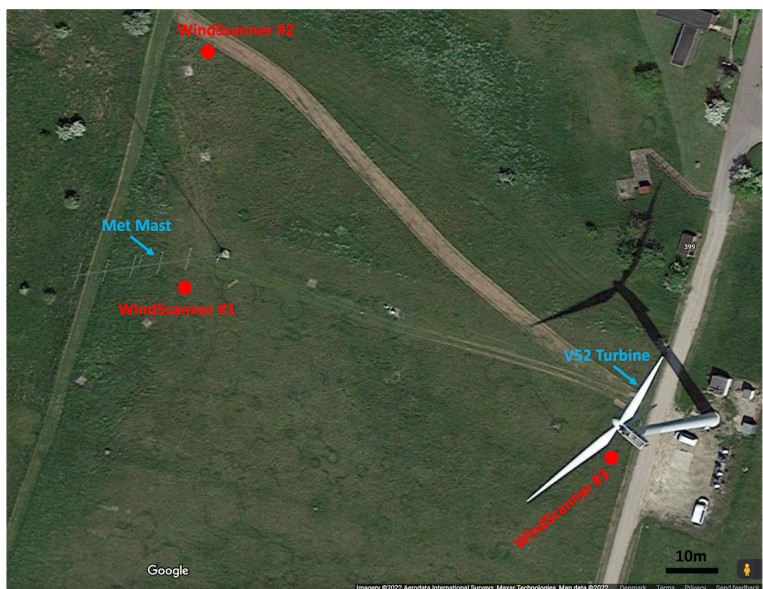

**Figure 1.** Top view of the locations of three WindScanner lidars (indicated by the red points) near the Vestas V52 wind turbine from © Google Maps.

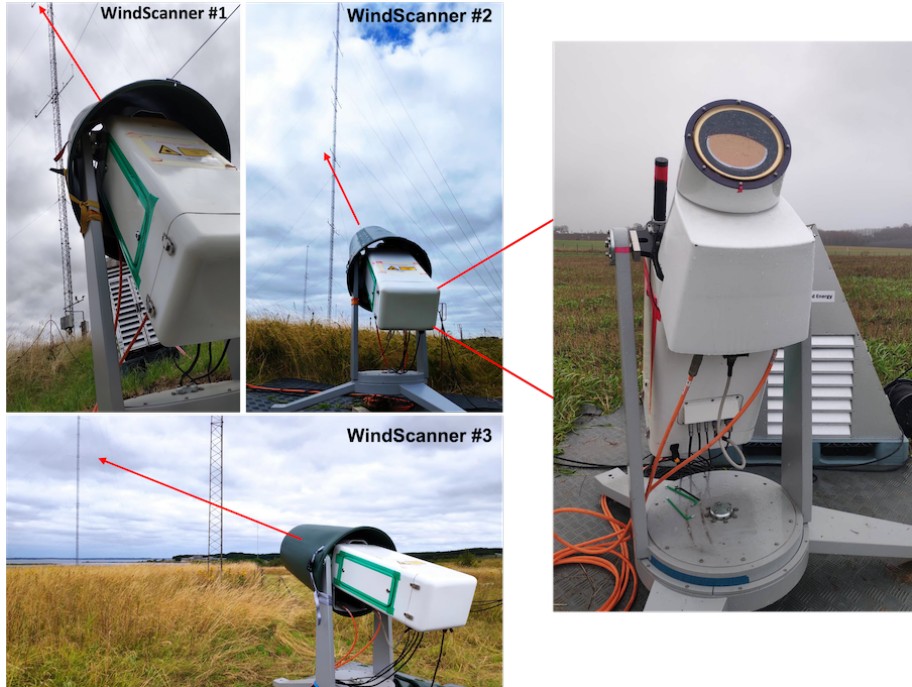

**Figure 2.** Three WindScanner lidars pointed at a common focus point close to a sonic anemometer on a met mast at DTU Risø campus.



**Table 1.** Summary of the measurement parameters of the WindScanner lidars.

|  | Elevation angle (°) | Rayleigh length $z_R$ (m) | Line-of-sight focus distance $R$ (m) | Angle to the North (°) |
|---|---|---|---|---|
| WindScanner #1 | 57.9 | 0.6 | 37.2 | 222.6 |
| WindScanner #2 | 34.6 | 1.4 | 54.8 | -7.1 |
| WindScanner #3 | 15.3 | 4.9 | 103.9 | 119.3 |

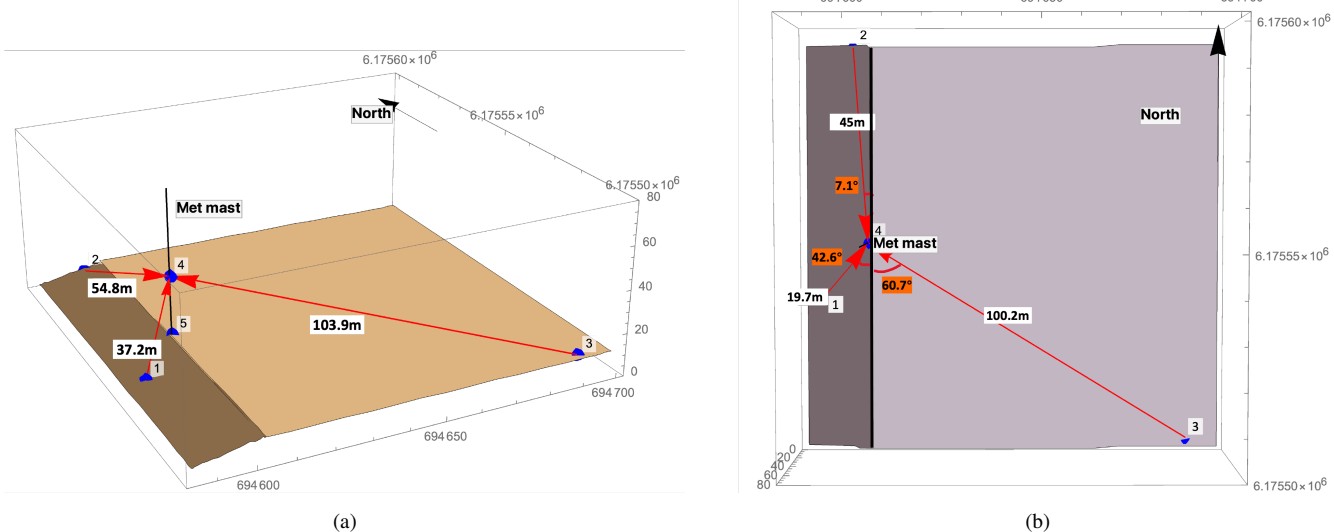

(a)                                   (b)

**Figure 3.** The 3D **(a)** and top view **(b)** of the configuration of the instruments at DTU Risø campus. The blue points marked by 1, 2, 3 are the three WindScanner lidars, focused at the point 4 which is 1 m north of a sonic anemometer at a height of 31 m above the ground. Point 5 is the base of the met mast. The black solid line indicates the met mast in **(a)**. The distances in **(a)** are the line-of-sight focus distances, while those in **(b)** are the projections on the horizontal plane.

The measurement period of the three WindScanner lidars is from 15:12 (UTC+1) to 23:29 (UTC+1) on September 27[th], 2022. All times mentioned in the paper are UTC+1.

The backscattered light is sampled at a rate of 120 MHz and Doppler spectra containing 512 frequency bins are calculated

with a corresponding wind speed spectral resolution about $0.183\ \mathrm{ms^{-1}}$. In order to be able to determine the sign of the line-of-sight velocity, the in-phase/quadrature-phase (IQ) detection method (Abari et al., 2014) is employed, which mixes the received signal with two local oscillator (LO) signals phase shifted by $90°$ relative to each other. Subsequently, a block averaging of 78 spectra takes place resulting in a final sampling rate of 3 kHz. Therefore, at every minute each WindScanner lidar will provide a data file containing 180000 spectra in total. The reason for setting the spectral sampling frequency to 3 kHz is that

the sampling period for a spectrum (0.33 ms) is less than the beam transit time (0.35 ms) of a typical rain drop. Here we found





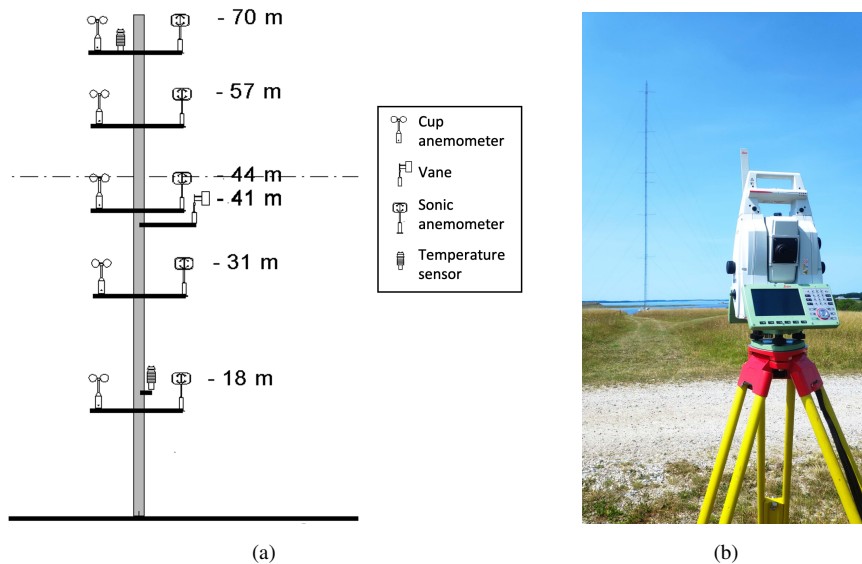

(a)                                          (b)

**Figure 4.** A sketch of the V52 meteorological mast with the instrumentation **(a)** and the Leica Total Station (Leica Geosystems, last access: 12 March 2023.) is used to scan the sonic anemometer at 31 m high **(b)**. The dashed line in **(a)** indicates the hub height of the DTU V52 wind turbine.

that the maximum downfall speed of raindrops is $9\,\mathrm{ms}^{-1}$ from the disdrometer measurement (Fig. 9b) and the calculated beam width is 3.14 mm when the focus distance is 103.9 m. Consequently, the rare instances where a raindrop resides in the beam could be identified and suppressed.

## 2.2 METEK sonic anemometer

The meteorological mast location is approximately 120 m northwest of the DTU V52 wind turbine and its base is 7.3 m above the sea surface (Fig. 1). There are five sonic anemometers (USA-1, Metek) on booms facing north and five cup anemometers (Risø) on booms facing south, placed at 18 m, 31 m, 44 m, 57 m, and 70 m above the terrain (Fig. 4). The sampling frequency of the sonic anemometers was 50 Hz. Furthermore, the mast is instrumented with a wind vane at 41 m, and two absolute temperature sensors mounted at 18 m and 70 m, respectively. In order to test the consistency of the mast wind measurements,

the available sonic and cup observations at different heights are compared in the following section. The sonic anemometer at 31 m is used as a reference for the further comparison with the radial wind velocity detected by the three WindScanner lidars.

In this step, it is also important to get accurate orientation of the sonic anemometer. For this purpose, the azimuth angle of the boom is considered as the direction offset of the sonic anemometer relative to the North. Here the Leica Total Station (Fig. 4b) was used to scan the sonic anemometer at 31 m height, the boom at the same height and the three WindScanner lidars.

Examples of the scanned results are presented in Fig. 5 and the azimuth angle of the boom to the north is $13.2°$ in UTM32 zone and the tilt angle of the sonic to the vertical is $1.9°$, which will be used to compute the unit vectors when projecting the



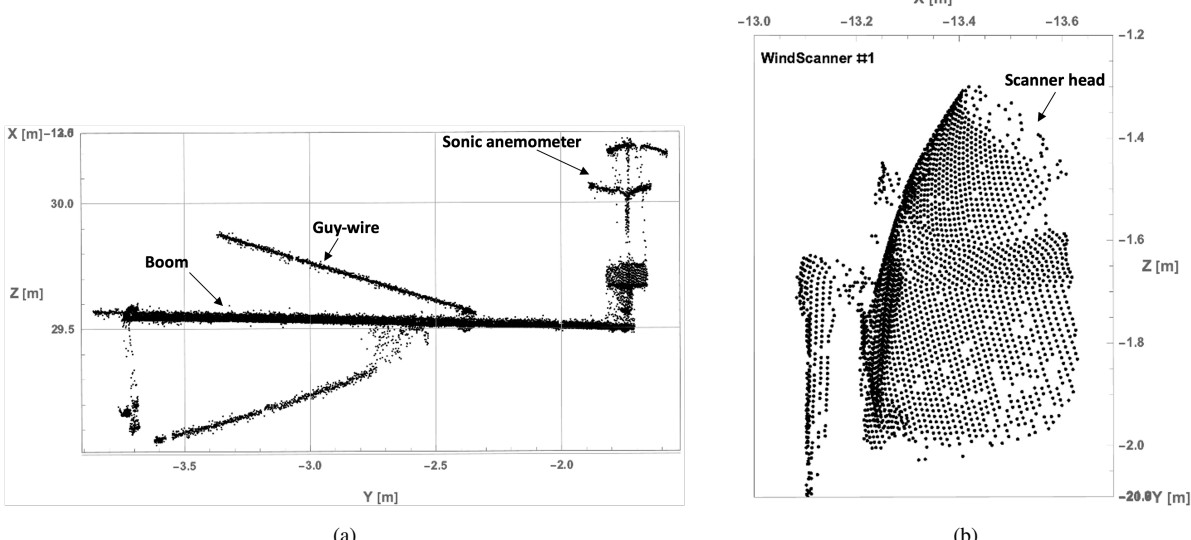

**Figure 5.** Point clouds of **(a)** the sonic anemometer at the height of 31 m and **(b)** the head of the WindScanner unit #1, plotted in the local coordinate system of the Leica Total Station. The scans were used to derive the unit vectors of the three lidar beams.

reference sonic velocity onto the directions of the three lidar beams. Consequently, the unit vectors of three lidars beams are $[-0.36, -0.39, -0.85]$, $[-0.10, +0.82, -0.57]$ and $[+0.84, -0.47, -0.26]$, respectively.

### 2.3 Disdrometer measurements

The falling velocity and diameter of the rain droplets were measured by a laser optical disdrometer manufactured by Thies (Laser Precipitation Monitor, LPM), with a transmitter head emitting a horizontal laser-light plane and a receiver head detecting the emitted laser light (Fig. 6). When a rain droplet intersects the laser beam, it attenuates the power of the transmitted laser light with a specific magnitude as a function of the falling velocity and the diameter. After the application of a proprietary algorithm, the measured droplets are classified to specific velocity and diameter classes, which are outputted with a temporal 125 resolution of 1 minute. Here, the droplet diameter is given as the equi-volume sphere diameter (Angulo-Martínez et al., 2018). Some technical details of Thies LPM disdrometer are given in Table 2.

### 3 Sonic anemometer and disdrometer data

### 3.1 10-minute averaged sonic data

Before analysing the sonic and lidar data, the sonic and cup wind speed as well as the sonic and vane wind direction at different 130 heights were compared. In Fig. 7 we show that the 10-minute averaged wind speeds by sonic and cup anemometers are in good agreement for all heights including the height of 31 m where the lidars were measuring. The slope of a linear regression is



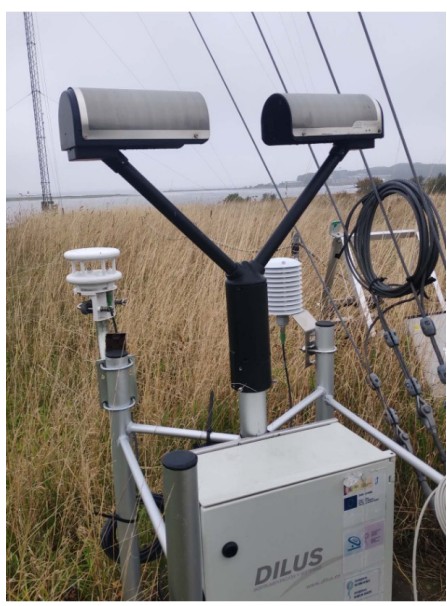

**Figure 6.** Thies Laser Precipitation Monitor(LPM) at DTU Risø campus.

**Table 2.** Technical detailed of the Thies Laser Precipitation Monitor (LPM).

|  | Thies LPM |
| --- | --- |
| Laser wavelength [nm] | 786 |
| Size of laser-light plane [mm$^2$] | 4560 |
| Number of diameter classes | 22 |
| Min/max of diameter classes [mm] | $0.1875/ \geq 8$ |
| Number of velocity classes | 20 |
| Min/max of velocity classes [ms$^{-1}$] | 0.1/15 |

1.008 with a coefficient of determination $R^2$ equal to 0.997, which shows that wind speeds measured by the sonic anemometers agree well with that measured by cup anemometers (with only a 1% difference). The same conclusion can be drawn for the wind direction in Fig. 8. Besides, the mean absolute difference of wind speed between the sonic and cup anemometer at 31 m height is 0.11 ms$^{-1}$ and that for wind direction between the sonic at 44 m and the vane at 41 m height is 1°. However, for further comparison with the lidar data, the three unit vectors describing the direction of the line-of-sight are used to project the wind vector measured by sonic anemometer onto the lidar's line-of-sight, as mentioned in Sect. 2.2.





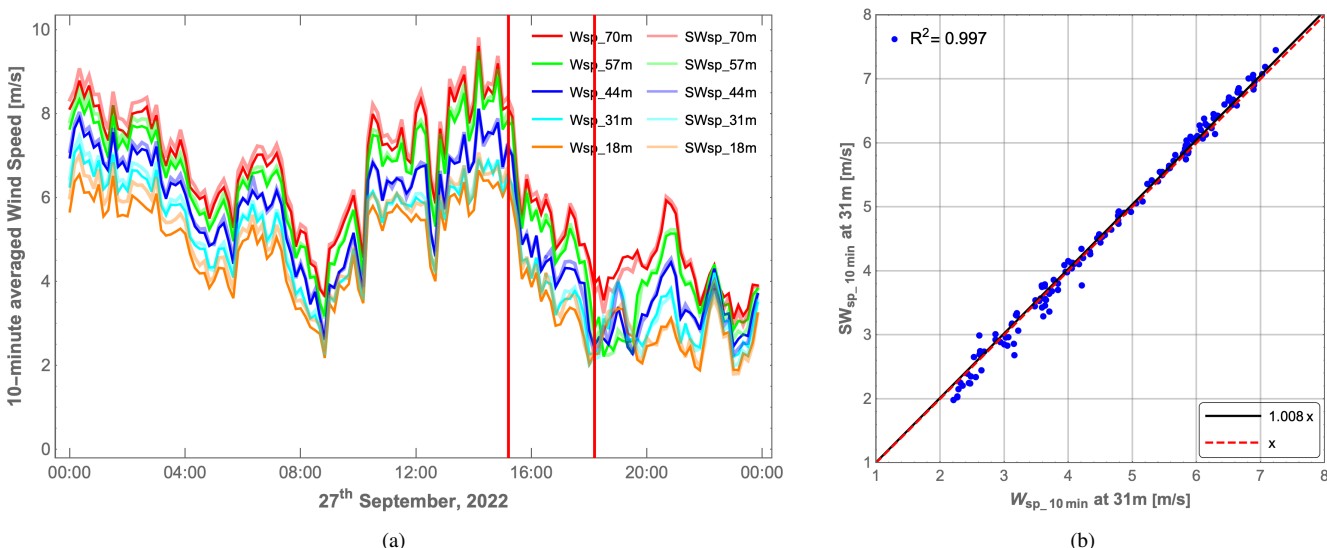

(a)            (b)

**Figure 7.** Comparison of wind speed measured by sonic and cup anemometers at five vertical heights (18, 31, 44, 57 and 70 m) in **(a)** and the linear regression between sonic and cup wind speed as well as the fitted slope in **(b)**. "$W_{sp}$" stands for the wind speed by cup anemometer, while "$SW_{sp}$" means measured by sonic anemometer. The two red lines mark the comparison period of lidar and sonic data from 15:12 to 18:11.

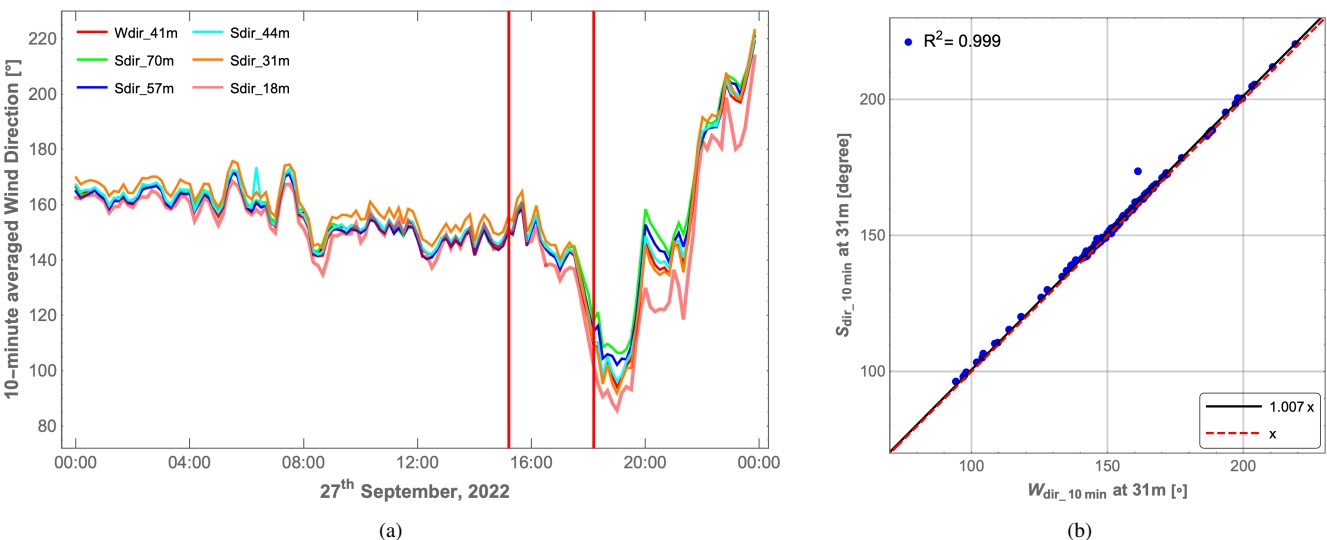

(a)            (b)

**Figure 8.** Comparison of wind direction measured by sonic anemometers at five vertical heights (18, 31, 44, 57 and 70 m) and by wind vane at 41 m in **(a)** and the linear regression between sonic and vane wind direction as well as the fitted slope in **(b)**. "$W_{dir}$" stands for the wind direction by vane, while "$S_{dir}$" means measured by sonic anemometer. The two red lines mark the comparison period of lidar and sonic data from 15:12 to 18:11.



Considering the scanners started to measure from 15:12, the rain stopped after 19:00, and the wake influence of the DTU V52 wind turbine, three full hours of data was selected from 15:12 to 18:11 for the comparison, which is marked by the two red vertical lines in Fig. 7 and Fig. 8. From 15:12 to 18:11, the 10-minutes mean wind speed by sonic anemometer at 31m is in the interval $[2.02 \text{ ms}^{-1}, 6.59 \text{ ms}^{-1}]$, while the wind direction is in the interval $[110.9°, 164.8°]$.

## 3.2 1-minute disdrometer data

The 1-minute averaged rain intensity from 15:00 to 19:30 measured by Thies disdrometer is shown in Fig. 9a. It started to rain from 15:15, reached the highest precipitation rate about $4 \text{ mmh}^{-1}$ at 15:48 and stopped after 19:00. The Met Office defines moderate rain as a precipitation rate between 2.5 mm and 7.6 mm per hour. The selected comparison period from 15:12 to 18:11 includes no-rain, light-rain (the precipitation rate is smaller than $2.5 \text{ mmh}^{-1}$) and moderate-rain minutes, which enables the investigation of the performance of the method we propose to filter away rain signals during precipitation levels. During the highest rain intensity period, most of the rain drops have the mass-weighted mean diameters smaller than 2 mm and the falling velocity smaller than $6 \text{ ms}^{-1}$ as shown in Fig. 9b.

## 4 Suppression method of the rain-bias

### 4.1 Lidar data processing

The raw lidar data without filtering away the rain signal (marked by "Raw") is processed by following the conventional procedure to retrieve a line-of-sight wind velocity. Firstly, every raw spectrum (the blue curve in Fig. 10a) is divided with the background spectrum (the red curve in Fig. 10a) to flatten the noise floor. The background spectrum is calculated as the median power spectral density per frequency of 180000 Doppler spectra, acquired during a one-minute period. After that, we choose the smaller background for any pair of frequencies $(-f, f)$, which provides the true background even if the wind velocity is constant over the minute. However, if the wind velocity is around zero, this procedure does not work. Therefore, in the case of lidar #1 where the line-of-sight velocity fluctuates around zero, a background spectrum is calculated for a period where the line-of-sight speed is away from zero.

Secondly, the 3 kHz spectra are averaged down to 50 Hz to ease comparison with the sonic anemometer. Then, the spectral threshold (the red line in Fig. 10c) of each spectrum is calculated based on the mean value ($\mu$) plus a multiple number of the standard deviation ($\sigma$) of the power spectral density over a wind-free Doppler frequency range. The final 50 Hz spectrum used to estimate the line-of-sight wind velocity is obtained from Fig. 10c, by subtracting the spectral threshold and zeroing the negative values. Three different methods can typically be used to find the line-of-sight velocity from the spectrum: the maximum method, the centroid method, and the median method (see comparison made by Held and Mann, 2018). The median method is employed in this investigation because it is characterized by the least biases in the case of weak signals (Angelou et al., 2012a).



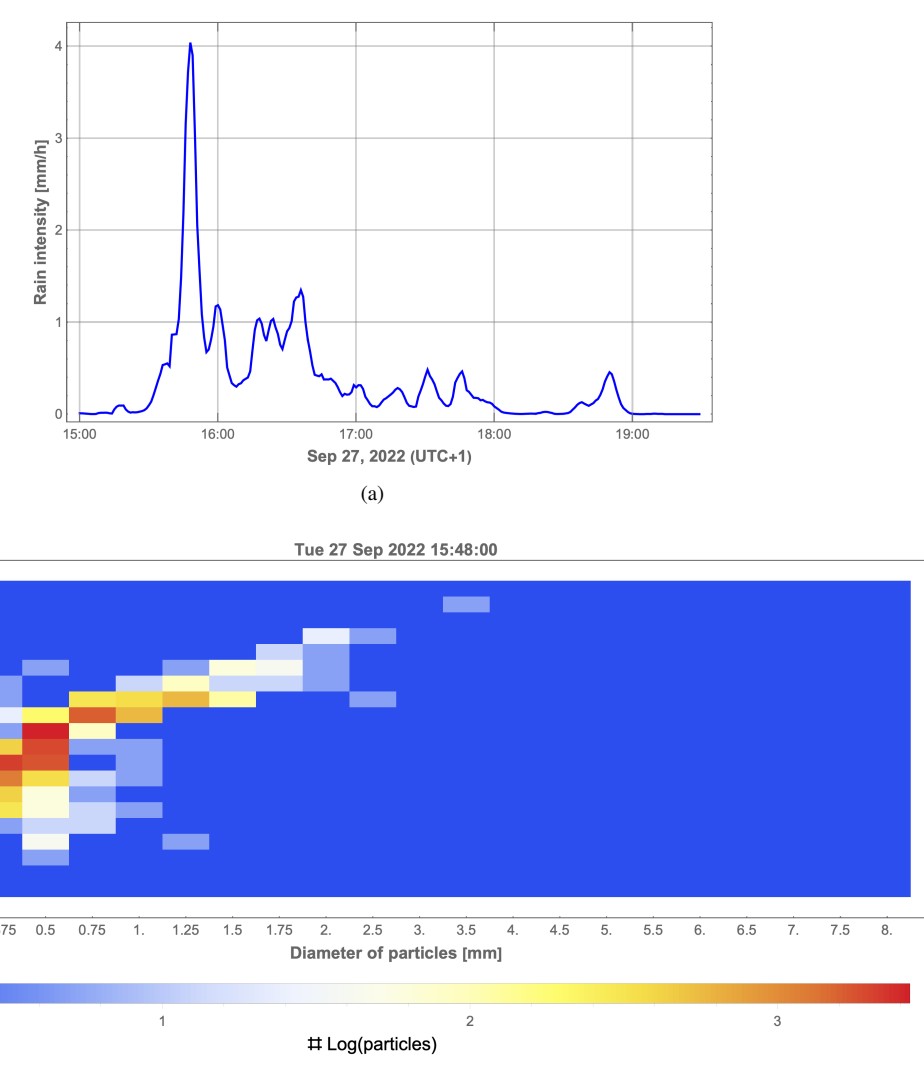

**Figure 9.** Rain intensity from 15:00 to 19:30 on September $27^{th}$, 2022 based on the measurement from the Thies Laser Precipitation Monitor disdrometer **(a)** and the color-coded boxes represent the distribution of the number of measurements with specific vertical falling speeds and mass-weighted mean diameters at one minute (15:48) with the highest rain intensity **(b)**.





However, from a random 3 kHz spectrum acquired during a 1-minute period (15:48) with a moderate-rain precipitation, it is obvious that sometimes the spectrum has a very high, narrow peak as shown in Fig. 10b. This is caused by a raindrop falling
through the beam, the intensity of which should be compared to the ones of the more commonly occurring spectra where the Doppler signal is caused by the aerosols (Fig. 10a). Here the width of the Doppler spectrum in Fig. 10a is relatively wider because the aerosols within the measurement volume of the lidar have slightly different velocities and the peak value is much lower. In contrast, the spectrum caused by the rain drop is very narrow because of the single velocity of the drop. From the histogram (Fig. 10d) of the maximum values of the spectra obtained during this moderate-rain minute, the very high back-
scattering events marked by the red circle are the large raindrops passing through the center line of the laser beam close to the beam waist. These could potentially cause a bias between the radial wind velocity measured by the lidar and by the sonic.

Therefore, based on the above observations, a method is proposed to reduce the influence of the rain on the averaged Doppler spectra. This is done by normalizing each individual noise-flattened spectrum with its peak value and then down sample the 3 kHz spectra to 50 Hz. The subsequent steps would be the same as the aforementioned process to deal with the original raw
spectra, i.e., subtract the spectral threshold, replace negative values by zero and lastly apply the median velocity estimation method to calculate wind velocity.

After flattening the noise floor, it is vital to determine a correct spectral threshold to define the signal caused by the wind in a Doppler spectrum. This is because a too high spectral threshold would result in the unexpected removal of the useful Doppler signal and causing false $0 \text{ ms}^{-1}$ wind velocity, while a too low spectral threshold would leave a lot of noise in the spectrum,
deteriorating the accuracy of the wind velocity estimation. As concluded in (Angelou et al., 2012a), the optimum number of standard deviations for defining the threshold is not the same for different data sets and a number of 2.5 has been used for the three lidars in this investigation.

### 4.2   Lidar spectra with and without rain-removal normalization

It is important to point out that during the measurement time from 15:12 to 18:11, WindScanner #2 was facing the wind almost
all the time and we speculate that rain water was covering the entrance window of the lidar telescope, despite our attempt to shield the window with a green barrel (Fig. 2). The water caused a very weak Doppler spectrum even at the minute with the highest rain intensity. Therefore, for further analysis and comparison, only the measurement data by lidar #1 and #3 are used.

It is worth noting that the wind direction at the minute with the highest rain intensity (15:48) is from $160°$ by the 10-minute averaged sonic data and the orientations of the two scanners are $222.6°$ and $119.3°$ to the north (Fig. 3b), revealing that at this
minute the wind moves away from the laser beams of both lidars, causing red Doppler shift. Consequently, the projection of the resultant velocity of raindrops, in the measuring configuration used here, is smaller than that of the horizontal wind speed onto the beam direction. In Fig. 11, it is very obvious that after normalization by the spectral peak, the narrow Doppler signal caused by the raindrops (marked by the red arrow) is effectively suppressed and the bias between the reference sonic wind velocity and that of the lidars is reduced as can also be seen in Table 3, for example, from $-1.56$ to $-0.18 \text{ ms}^{-1}$ at 50 Hz for
lidar #3. This indicates that normalization by the spectral peak value can help to reduce the influence of the rain droplets since the narrow peak closer to the center zero frequency (the solid black line) is strongly suppressed.





**Figure 10.** Examples of a normal Doppler spectrum containing wind signal **(a)** and a spectrum containing rain signal **(b)** at one moderate-rain minute (15:48) with the highest rain intensity. **(c)** is one example of a 50 Hz noise-flattened spectrum and the corresponding spectral threshold. **(d)** shows the histogram of the maximum power spectral density values $S_{max}$ of 180000 raw spectra over the duration of the same minute. The red curve in **(a)** and **(b)** is the mean background spectrum of this one minute. Toward the right of **(d)**, a red circle marks the strongest rain signals.

**Figure 11.** Comparison of 50 Hz ((**a**), (**b**)) and 1 Hz ((**c**), (**d**)) spectra containing both wind and rain signal for WindScanner lidar #1 and #3 at one moderate-rain minute (15:48). The red solid curves represent the standard-average spectra without filtering away the rain signals (marked by "Raw"), while the blues are after rain-suppressing normalization by the spectral peak to filter away the rain signals (marked by "Norm" short for normalization). The red and blue dashed lines represent for the median frequency bin of the standard averaged and the rain-suppressing averaged Doppler spectra, which are used to derive line-of-sight wind velocity. The green dashed line indicates the sonic wind velocity. The solid black line stands for the zero frequency bin.



**Table 3.** The estimated wind velocity by lidar data with ($V_{norm}$) and without ($V_{raw}$) normalization, and by the sonic anemometer ($V_{sonic}$) from 50 Hz and 1 Hz spectra at one moderate-rain minute (15:48).

|  | $V_{sonic}$ (ms$^{-1}$) | $V_{raw}$ (ms$^{-1}$) | $V_{norm}$ (ms$^{-1}$) |
|---|---|---|---|
| 50 Hz of #1 | -1.67 | -1.06 | -1.62 |
| 50 Hz of #3 | -4.58 | -3.02 | -4.40 |
| 1 Hz of #1 | -1.14 | -0.50 | -1.10 |
| 1 Hz of #3 | -4.72 | -4.29 | -4.59 |

Therefore, based on the promising results about the effective suppression of the rain Doppler signals at one moderate-rain minute (15:48), in the following section, we compare the radial wind velocity detected by WindScanner lidars and the sonic anemometer at 31 m height in details. The outcomes are elaborated to verify this rain-suppressing normalization method under no-rain, light-rain and moderate-rain conditions.

## 5 Comparison between lidar and sonic wind velocity

### 5.1 50Hz wind velocity comparison

The reference 50 Hz sonic data at 31 m height was synchronized with the lidar measurements before the comparison. Based on the sonic status information, we interpolated repeated sonic wind velocity measurements, that occurred due to the influence of the raindrops. In Fig. 12a, c, e and Fig. 13a, c, e, the 50 Hz radial wind velocity time series of the normalized lidar data (the blue curves) matches well with the synchronized sonic data (the green curves) at the no-rain, light-rain ($I_{rain} = 1$ mmh$^{-1}$) and moderate-rain ($I_{rain} = 4$ mmh$^{-1}$) minutes. It is very clear that the fluctuation of the wind velocity caused by the raindrops is effectively suppressed, especially during the moderate-rain period for lidar #1 with a shorter focus distance 37.2 m in Fig. 12e, or during the rainy period for lidar #3 with a longer focus distance 103.9 m in Fig. 13c and e. This can also be found from $R^2$ of the scatter plots in Fig. 12 and 13, indicating less dispersion of the lidar wind velocity after rain-suppressing normalization.

Furthermore, Tables 4 and 5 compare the minute-averaged radial wind velocity of the three data sets (sonic, standard-averaged lidar data, and rain-suppressing normalized lidar data) as well as the bias between the sonic and lidar estimations. In the case of small probe lengths (lidar #1), only at the moderate-rain minute, the bias is effectively reduced from $-0.15$ to $-0.04$ ms$^{-1}$ after normalization, whereas the bias is almost the same at the no-rain and light-rain minutes. However, precise wind velocity is obtained after normalization of lidar #3 data in the presence of light rain and moderate rain, with the bias correspondingly reduced from $-0.21$ to $-0.01$ ms$^{-1}$ and from $-0.33$ to $-0.08$ ms$^{-1}$. In light of this, it follows that when the probe length is small and it rains heavily than lightly, rain-suppressing normalization by the spectral peak value can filter out the rain signals effectively. However, when the probe length is larger (up to 10 m) with a broader Lorentzian weighting





**Table 4.** 1-minute averaged wind velocity based on 50 Hz data and the corresponding bias between the sonic anemometer and WindScanner lidar #1 at three minutes, with (norm) and without (raw) normalization.

| | $V_{sonic}$ (ms$^{-1}$) | $V_{raw}$ (ms$^{-1}$) | $V_{sonic} - V_{raw}$ (ms$^{-1}$) | $V_{norm}$ (ms$^{-1}$) | $V_{sonic} - V_{norm}$ (ms$^{-1}$) |
|---|---|---|---|---|---|
| No-rain minute 15:13 | -1.01 | -1.07 | 0.06 | -1.08 | 0.07 |
| Light-rain minute 16:36 | -0.38 | -0.391 | 0.11 | -0.394 | 0.14 |
| Moderate-rain minute 15:48 | -0.64 | -0.49 | -0.15 | -0.60 | -0.04 |

**Table 5.** 1-minute averaged wind velocity based on 50 Hz data and the corresponding bias between the sonic anemometer and WindScanner lidar #3 at three minutes, with (norm) and without (raw) normalization.

| | $V_{sonic}$ (ms$^{-1}$) | $V_{raw}$ (ms$^{-1}$) | $V_{sonic} - V_{raw}$ (ms$^{-1}$) | $V_{norm}$ (ms$^{-1}$) | $V_{sonic} - V_{norm}$ (ms$^{-1}$) |
|---|---|---|---|---|---|
| No-rain minute 15:13 | -5.42 | -5.41 | -0.01 | -5.45 | 0.03 |
| Light-rain minute 16:36 | -3.37 | -3.16 | -0.21 | -3.36 | -0.01 |
| Moderate-rain minute 15:48 | -3.62 | -3.29 | -0.33 | -3.54 | -0.08 |

function, normalization performs very well when rain falls (whether light or heavy) because of the sensitivity of the lidar to
rain signals.

In addition, the same conclusions can be drawn by comparing the probability density function (PDF) calculated for the radial wind velocity estimated based on the 1-minute averaged lidar spectra and the 50 Hz sonic data at three minutes, as shown in Fig. 14. The improvement by normalization for lidar #1 with a smaller probe length is observed only during the moderate rain period (Fig. 14e), as the calculated integral of the absolute difference of the PDF is reduced from 3.04 to 1.08 in Fig. 15. For
lidar #3 with a larger probe length, normalization performs very well not only at the moderate-rain minute in Fig. 14f, but also at the light-rain minute in Fig. 14d with the reduction of the integral of the absolute difference of the PDF from 1.68 to 0.57. In the comparison of the integral value of the absolute difference of the PDF alone, normalization performs very well during rain periods when the probe length is large, or during the moderate rain when the probe length is smaller, which is consistent with the conclusions discussed above.

When comparing $R^2$ of the original raw lidar data at the same minute between lidar #1 in Fig. 12b, d, f and lidar #3 in Fig. 13b, d, f, it is easy to find that $R^2$ of lidar #3 is smaller than that of lidar #1 at every minute. This is because the larger probe length of lidar #3, 9.8 m, dominantly influences the variation of the estimated wind velocity, even though the elevation angle of lidar #1 (57.9°) is much larger than that of lidar #3 (15.3°). Besides, due to the very big elevation angle of lidar





**Figure 12.** Comparison of 50 Hz radial wind velocity (the left column) and the scatter plot (the right column) of the sonic data (the green curves), the raw (the red curves and dots) and the normalized lidar data (the blue curves and dots) for WindScanner lidar #1 at the no-rain, light-rain and moderate-rain minutes from top to bottom. The solid black line indicates y=x.





**Figure 13.** Comparison of 50 Hz radial wind velocity (the left column) and the scatter plot (the right column) of the sonic data (the green curves), the raw (the red curves and dots) and the normalized lidar data (the blue curves and dots) for WindScanner lidar #3 at the no-rain, light-rain and moderate-rain minutes from top to bottom. The solid black line indicates y=x.





**Figure 14.** Comparison of the probability density function (PDF) of the estimated radial wind velocity by 1-minute averaged spectra of the raw lidar data without normalization (the red curves) and the normalized lidar data (the blue curves) as well as the histogram of the 50 Hz sonic data at the no-rain (top row), light-rain (second row) and moderate-rain (bottom row) for WindScanner lidar #1 (**(a)**, **(c)** and **(e)**) and #3 (**(b)**, **(d)** and **(f)**).





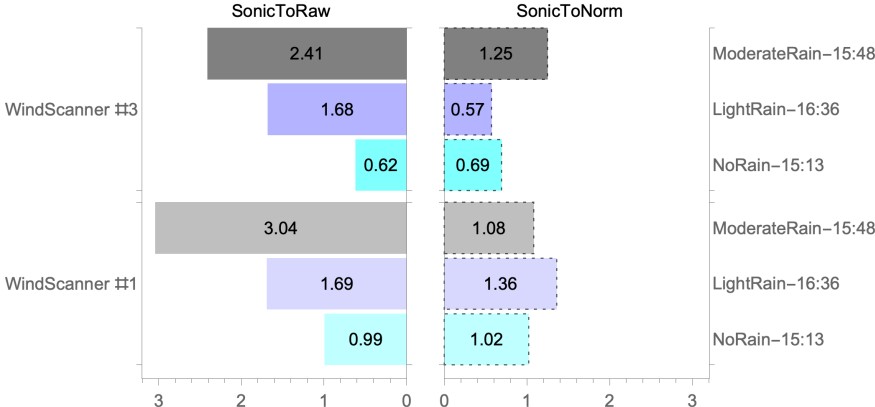

**Figure 15.** Comparison of the integral value of the absolute difference of the PDF between the sonic and the lidar data sets with (the right column) and without (the left column) rain-suppressing normalization at no-rain, light-rain and moderate-rain minutes of the two WindScanners based on the probability density function (PDF) in Fig. 14.

#1, it would be easier to have a smaller projection of the horizontal wind speed or of the resultant raindrop velocity on the
line-of-sight beam direction, even zero projection at some cases, as depicted in the scatter plots in Fig. 12b, d and f.

## 5.2 1-minute wind velocity comparison

The bias between 1-minute sonic wind velocity and the lidar wind velocity, along with the rain intensity, are presented in Fig. 16. From the figure we can draw similar conclusions as described previously. In the case of the lidar #1 with a smaller probe length in Fig. 16a, after normalization with the spectral peak, the large bias of the red curve around the rain intensity
peak is effectively reduced from $-0.15$ to $-0.03$ ms$^{-1}$. This is a result of filtering away the low and negative velocity caused by raindrops. For other minutes, the estimated wind velocity after normalization is almost the same as the raw data, which is in line with the conclusions from 50 Hz data in Sect. 5.1.

For lidar #3, the improvement of wind velocity estimation by normalization is highly effective as presented in Fig. 16b, from when it started to rain at 15:29 until 16:48. Afterwards at some minutes, the wind velocity time series after normalization
overlaps with that of the raw lidar data, especially when the rain intensity is below $0.2$ mmh$^{-1}$ after 17:00. For most of the three-hour comparison period, the wind velocity calculated by the raw lidar data is underestimated, as shown in Fig. 16b. This is because of the small projection of the raindrop velocity, which counteracts the aerosol projection and adversely affects the estimated wind velocity. As well, the red curve shows a radial velocity difference of over $0.5$ ms$^{-1}$.

In Fig. 17c, the 1-minute lidar wind velocity after the rain-suppressing normalization matches well with that of the sonic
measurement for lidar #3 with a larger probe length, as the filtered lidar data (blue dots) are in a closer agreement with the sonic measurements compared with the raw lidar data (red dots). For lidar #1 in Fig. 17a, there is no obvious improvement after normalization by the spectral peak. However, the averaged bias in Fig. 17b and d demonstrates the performance of rain-suppressing normalization. It is clearly indicated by the red and blue fitted curves that the suppression becomes effective not



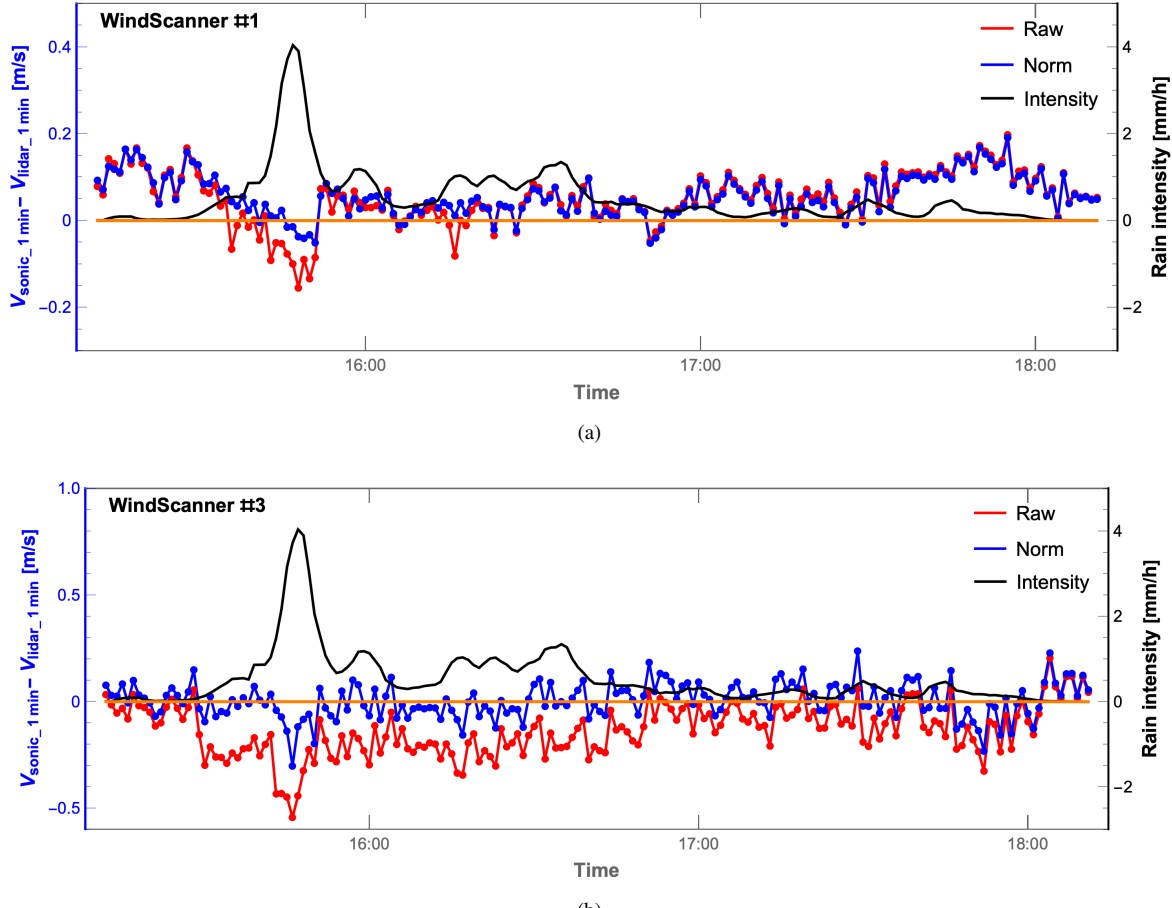

**Figure 16.** Difference of 1-minute averaged radial wind velocity between the WindScanner lidars and the sonic anemometer data together with the rain intensity (the solid black curve) from 15:12 to 18:11 for lidar #1 ((**a**)) and #3 ((**b**)). The raw lidar data is marked in red, while the blue curve and dots represent for the normalized lidar data.

only for lidar #3 when it rains, but also for lidar #1 with a short focus distance when the rain intensity $I_{rain}$ is large. We

speculate that during light rain, the choice of this method does not have a large impact on the determined velocities of lidar #1 due to the rare occurrence of raindrops passing through the laser beam with a very short probe length. These lead to the same conclusions discussed previously that rain-suppressing normalization performs well for the large probe length when it rains as well as for the small probe length when it rains more than lightly.

## 6   Conclusions

By sampling the spectra of a Doppler lidar faster than the raindrop's beam transit time, the rain signal can be filtered away and the bias on the wind velocity estimation can be reduced. This is verified by sampling lidar spectra at 3 kHz with different





**Figure 17.** The scatter plot of 1-minute wind velocity time series ((**a**), (**c**)) and the averaged bias between the lidar and sonic measurements as a function of the rain intensity ((**b**), (**d**)) from 15:12 to 18:11 for WindScanner lidar #1 (top row) and #3 (bottom row). The red and blue curves in (**b**) and (**d**) represent the fitted function of the bias.

elevation angles and focus distances, and by comparing with a reference sonic anemometer. In the method we propose, $3\,\text{kHz}$ spectra are normalized with their peak values before they are averaged down to $50\,\text{Hz}$ from which the radial wind velocity is determined. Over the whole range of rain intensities, we have observed a significant reduction of the bias of the lidar measurements relative to the sonic. The tendency is that the more it rains, the more the bias is reduced. For moderate rain intensity, the range of the bias is reduced from the interval $0.1$ to $0.4\,\text{ms}^{-1}$ to $0.0$ to $0.1\,\text{ms}^{-1}$.

*Data availability.* Data underlying the results presented in this paper can be obtained from the authors upon reasonable request.

*Author contributions.* All authors have made a contribution to the paper preparation. Conceptualization, JM, LJ, NA, MS; methodology, project management and experiment conduction, LJ, NA, JM, MS; data analysis, LJ, JM; writing—original draft preparation, LJ; writing—review and editing, LJ, NA, JM, MS. All authors have read and agreed to the published version of the manuscript.

*Competing interests.* The authors declare no conflict of interest.

*Acknowledgements.* The authors are grateful to senior scientist Gunner Chr. Larsen (DTU) for his continued support and inspiration for the project. The authors would also like to thank Michael Courtney, Ebba Dellwik, Per Hansen, Karen Enevoldsen, Lars Christensen, Michael Rasmussen, and Claus Brian Munk Pedersen from DTU, for their helpful support during the field experiment, fruitful discussions and helpful comments.

*Financial support.* This research is mainly funded by the Innovation Training Network Marie Skłodowska-Curie Actions LIKE (LIdar Knowledge Europe) project and the LICOREIM (LIdar-assited COntrol for REliability IMprovement) project. The LIKE project (H2020-MSCA-ITN-2019, Grant number 858358) is funded by the European Union. The LICOREIM project (Grant number 64019-0580) is funded by the Energy Technology Development and Demonstration Program (EUDP).

*Review statement.* This paper was edited by and reviewed by two anonymous referees.



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
