# Peer review of "Suppression of precipitation bias on wind velocity from continuous-wave Doppler lidars"

_EGUsphere, 2023_

## Referee Comment (RC2)

**Suppression of precipitation bias on wind velocity from continuous-wave Doppler lidars**

Liqin Jin, Jakob Mann, Nikolas Angelou, and Mikael Sjöholm

Paper ID: amt-2023-464

Iteration: First review round.

**REVIEW COMMENTS**

**OVERALL SUMMARY**

The authors address a timely subject in wind-lidar remote sensing, which is the removal of the bias caused by precipitation in the measurement of wind velocity. This topic is of interest aligned with different research programmes in the EU and US (e.g., NASA Clouds, Aerosol and Precipitation programme).

The authors successfully demonstrate the proposed bias-removal method by conducting a field experiment at the Risø of the Technical University of Denmark that uses three wind-scanner lidars staring at same point and reference sonic anemometer measurements. The experiment discusses the impact of different rain rates (although limited to a maximum of 4 mm/h) and –up to a point- the effect of different lidar focusing lengths. The paper is well written and easy to read. The experimental part was carefully planned and executed.

Sect. 4.1 addressing the signal processing method – a core part of the manuscript - could be improved a bit along the lines suggested by the General Comments below. Sect. 5 figures could be streamlined and summarised considering that the manuscript already contains as much as fifteen figures, many of them multi-panel.

All considered, I recommend minor revision along the lines suggested by the general comments below.

**GENERAL COMMENTS**

My first comment essentially addresses the processing methodology of Sect. 4.1 in light of making –an already good manuscript- into a better structured and self-contained manuscript to read. Please consider these comments:

- The signal processing part is a bit weak. Please include a processing diagram block describing: (i) the standard and (ii) the new signal processing method proposed. This is a core part of the paper. Please connect the contents of Sect. 4.1 with the block diagram.

- Structure: I suggest dividing Sect. 4.1 in three parts:

1) Standard signal processing of the 3 kHz Doppler spectra: (L152-L159) + L161-L167. I'd say L160 talking about the 50 Hz spectrum is orphan and should be moved somewhere close to L179 when you start talking about the down sampling of the spectrum. Clearly separate between 3-kHz and 50-kHz processing. Clearly enunciate the down-sampling block.

2) Comparison between aerosol and rain Doppler spectra (L168-176).

3) The "proposed" method of the paper (L177-187).

The support of literature references included in this section is weak.

Second, I think the amount of Figure panels in Sect. 5 could substantially be reduced or moved to a Supplementary Materials Section considering that the manuscript already contains as much as fifteen figures, many of them multi-panel. For example, Fig. 12 could be skipped and Fig. 13 retained along with summary comments given in the text or with the help of a supporting Table. Most of the scatter plots can substituted by a Table describing the determination coefficients obtained plus a link to the Appendix /

Supplementary Materials for the interested reader. Similarly, Fig. 14 could be streamlined by including only panels b-d-f (WindScanner #3) and a comment or Table with descriptive PDF statistics.

But I leave the final selection of Figures/panels to the authors, or alternatively, to choose a better art arrangement.

**SPECIFIC COMMENTS**

L5 "the noise-flattened Doppler spectra." Consider: "the noise-flattened 3-kHz-sampled Doppler spectra".

L9 Consider: "at 50-Hz (20 ms) temporal resolution"

L10 Please clarify "conventional"

L25-30 In Sect. Introduction please comment a bit on the probe-length turbulence averaging effects in comparison to e.g., cup anemometers since this is an important drawback of focusing lidars (e.g, averaging of spatial turbulence scales).

L26 Change "that" into "which"

L54 Not sure acronym "CW" (continuous wave) has previously been defined. Preferably, use "CW" in caps.

L84 "can be approximated as [REF needed],"

Tab. 4. Please check if "angle to the North" is computed correctly. In a Cartesian coordinate system, angles are defined positive CCW. And angles between vectors (or between e.g., vector "1" and vector North) are computed by using equipollent vectors so that their origin coincides with the Cartesian origin (i.e., point 5 = projection of point 4 in the XY plane). From the geometrical angles given in Fig.3b and assuming North is 0 deg then I'd say "Angle to North" should be: (WindScanner 1) -42.6 deg; (WindScanner 2) 180 + 7.1.deg = 187.1 deg; (WindScanner 3), 60.7 deg. Please clarify if other Math/Physics conventions are used.

L95a Please state and clarify to the reader the "key" numbers of the processing. Don't let the reader guess them. Specifically:

1) Fast Fourier Transform (FFT) frequency_resolution: 120 MHz / 512 samples = 234.4 kHz/samples → speed resolution = (lamdda/2)*freq_resolution = 0.183 m/s

2) Spectrum_estimation period = sampling rate (1/120 MHz) x 512 (samples/spectrum) x 78 spectrum/average = 332.8e-6 [s] → spectrum_estimation rate = 1/spectrum_estimation_period = 3 kHz

L95b Please briefly summarise which power spectral density (PSD) and which peak spectral estimation method is used to retrieve the Doppler shift. I think this should also be remembered to the reader and shortly discussed later on, in L153-155.

L93 I recommend to repeat "all times are UTC+1" in all figure captions involving time series to help the reader.

L100 "less than the beam transit time of a typical rain drop". Please add literature REFERENCE.

L101 Please briefly clarify how the 0.35 ms transit time was estimated. E.g., at 9 m/s fall velocity we get, 9 m/s x 3.14 mm = 28 ms (in the near field "waist" of the laser beam).

L109 Consider to give manufacturer/model for the wind vane and temp sensors.

L118 I recommend drawing X, Y and Z labels on Fig. 3a to help the reader identify the given unit vectors ([−0.36, −0.39, −0.85], [−0.10, +0.82, −0.57] and [+0.84, −0.47, −0.26]).

Tab. 2 What does symbol "/ ≥" means in "0.1875/ ≥ 8? Is that a typo?

Fig.7-8. Font size. The "10-min" text in X and Y labels of panels (b) can barely be seen. Please enlarge font and include in the captions "Comparison of 10-min wind speed".

Fig. 9 CAPTIONS (comment to be extended to all manuscript figure captions) I recommend setting label letters at the beginning of each sentence and not at the end or in the middle of the sentence, for better clarity. I recommend to begin each figure caption with a sentence giving an overview of what the figure is about. Then use follow-up labels (a), (b) addressing each panels. E.g., "Rain event September 27th, 2022, 15:00-19:30 measured by the Thies (…). **(a)** Rain Intensity. **(b)** Distribution of the number of measurements … (color coded)."

I also recommend including in the caption the temporal resolution of the data (although this may seem repetitive), e.g., 1 minute, in this case.

L155 Please introduce acronym "power spectral density (PSD)" See also comment L95b for discussion.

L157-160 Different issues. Unclear. I would suggest expanding the spectral estimation part. In detail:

a) "However, if the wind velocity is around zero, this procedure does not work." Why, taking into account that the noise PSD is not zero? Please clarify.

b) L158-160 "where the line-of-sight velocity fluctuates around zero". At this point in text, mention to the reader that vertical line at approx. bin 255 (please clarify bin no.) corresponds to the zero-Doppler shift.

c) Why such a negative red peak at bin 255 occurs for the background noise (red trace) in Fig. 10a?

L178 "down sample". Please clarify how the "down-sampling" procedure is carried out. Is it that given the normalised spectra, which make evident the rain returns as very high and narrow peaks, these spectra are screened out for "very large" peaks, therefore, removed from the spectra average? This is known as histogrammed filtering but no quantitative criterion is given. Please give a quantitative criterion for screening out the "rain returns" in Fig. 10d. For example, is that percentile 90 of the cumulative distribution? Please consider to include this histogrammed filtering block in the proposed processing diagram above.

Fig. 10. Please vertically align panels (a) and (c). Please use the same X-axis range to ease comparison.

Fig. 10 caption. Please add: "The solid black line stands for the zero frequency bin" (as in Fig. 11).

Fig. 10. Which method is used to compute the PSD? E.g. Periodogram or others. Please include literature reference.

Fig. 11 Caption, L4. "represent the median frequency bin" or "stand for the median frequency bin"

Fig. 11 (b)(d) Please use the same frequency-range (X-axis) in both panels.

L209 Change "that occurred" into "which occurred"

L222 "more" missing → it rains more heavily than lightly

Tabs. 4-5 Please repeat in caption key information on "rain intensity" and "probe length".

Tab. 4. Please stick to two decimal digits everywhere. Please check for errors/typos on column "light-rain minute"

Figs. 12-13. Legends: The logical order should be "sonic-raw-norm" instead of "raw-sonic-norm".

Fig. 16 caption. Please repeat in caption "probe length" values to help the reader.

CONCLUSIONS Please give conclusions on your findings about the performance of the methods for different probe lengths (short/large probe length, which is an important point –although more risky- or your research) as well as future lines. Part of the conclusions given in L183-185, L221-224, L244 should be rewritten / summarised in Sect. Conclusions.

---

## Author Comment (AC1)

**0.1 Response to Reviewer 1 Comments**

**Dear reviewer:**

We appreciate the time and effort that you have dedicated to providing your insightful comments on our paper. We have been able to incorporate changes to reflect all the suggestions provided by you. We have highlighted the changes within the manuscript.

**General comments:**

1 **Lidar #1 has a relatively bigger elevation angle of $57.9°$ compared to Lidar #3 of $15.3°$. Generally, the velocity difference between aerosols and raindrops appears in the vertical direction. Therefore, large elevation angles should suffer more influence from rain signals. While figure 17 exhibits the opposite results (Raw data with the red circle). The authors explain that the short probe length may contribute to it. I think adding a comparison experiment or detailed analysis will be better.**

Thank you for pointing this out. It's a good point. In this study, the two lidars have different focus distances and different elevation angles. We still need to investigate which factor matters more in the performance of our proposed rain-suppressing normalization method. Therefore, we reformulate the paragraph in **L254-257** as "At every minute, $R^2$ of lidar #3 is smaller than that of lidar #1 when comparing $R^2$ of the original raw lidar data in Fig. 10. We are uncertain about why rain seems to deteriorate the wind signal of lidar #3 more than that of lidar #1. It could have to do with the larger sample volume of #3 or the different elevation angles, but it could also have to do with a different amount of raindrops on the entrance windows of the telescope. The understanding of these sensitivities awaits more experimentation.".

2 **The proposed method is verified by continuous-wave Doppler lidar measurements. I'm also interested in whether it's also suitable for a pulsed Doppler lidar which often uses a collimated beam. The author is advised to add related discussions.**

Thank you for pointing this out. We agree with this comment. Therefore, we have added some outlooks regarding potential investigations with pulsed lidars and characterizing the rain in the conclusion part as "The suggested method in this study could also be investigated for rain events (containing heavy rain) on several days and also for pulsed Doppler lidars even though their measurement volume is quite larger than that of the continuous-wave lidars. Further investigations could also attempt to retrieve the falling velocity and the size distribution of raindrops using the fast Doppler spectra.".

---

## Author Comment (AC2)

**0.1 Response to Reviewer 2 Comments**

**Dear reviewer:**

We appreciate the time and effort that you have dedicated to providing your insightful comments on our paper. We have been able to incorporate changes to reflect most of the suggestions provided by you. We have highlighted the changes within the manuscript.

General comments:

1 My first comment essentially addresses the processing methodology of Sect. 4.1 in light of making an already good manuscript into a better structured and self-contained manuscript to read. Please consider these comments:

- The signal processing part is a bit weak. Please include a processing diagram block describing: (i) the standard and (ii) the new signal processing method proposed. This is a core part of the paper. Please connect the contents of Sect. 4.1 with the block diagram.

- Structure: I suggest dividing Sect. 4.1 in three parts:

  (a) Standard signal processing of the 3 kHz Doppler spectra: (L152-L159) + L161-L167. I'd say L160 talking about the 50 Hz spectrum is orphan and should be moved somewhere close to L179 when you start talking about the down sampling of the spectrum. Clearly separate between 3-kHz and 50-kHz processing. Clearly enunciate the down-sampling block.

  (b) Comparison between aerosol and rain Doppler spectra (L168-176).

  (c) The "proposed" method of the paper (L177-187).

The support of literature references included in this section is weak.

Thank you for pointing this out. We agree with this comment. We added a processing diagram block in the draft to show the spectral process steps of our proposed method. Now, Sect. 4.1 is restructured according to the suggestion. The first paragraph is about how Doppler spectra after being averaged to lower frequencies are processed. Then we show the comparison between normal Doppler spectra with only aerosol-induced Doppler signals and the spectra with rain-induced signals. Subsequently, we proposed our rain-suppressing normalization method. Besides, two more references are added in this section.

[Figure]

Figure 1: Processing block diagram of the rain-suppressing normalization method (the solid lines from ① to ③) to estimate wind velocity based on 3-kHz-sampled Doppler spectra. Doppler spectra at lower frequencies that do not resolve individual raindrops (like 50 Hz) are processed according to the purple path including the dashed purple line, ②, and ③.

2 Second, I think the amount of Figure panels in Sect. 5 could substantially be reduced or moved to a Supplementary Materials Section considering that the manuscript already contains as much as fifteen figures, many of them multi-panel. For example, Fig. 12 could be skipped and Fig. 13 retained along with summary comments given in the text or with the help of a supporting Table. Most of the scatter plots can substituted by a Table describing the determination coefficients obtained plus a link to the Appendix /Supplementary Materials for the interested reader. Similarly, Fig. 14 could be streamlined by including only panels b-d-f (WindScanner #3) and a comment or Table with descriptive PDF statistics.

But I leave the final selection of Figures/panels to the authors, or alternatively, to choose a better art arrangement.

Thank you for pointing this out. It would have been good to have a compacted paper. We agree with you. We have removed Fig. 1 due to its similarity to Fig. 3 and merged Fig. 4 and 5, Fig. 7 and 8, as well as Fig. 12 and 13. However, in our study, we have two lidars with different elevation angles and different focus distances. The rain-suppressing method we proposed has a different performance from the two lidars. Therefore, we would like to keep Fig. 12 (now merged in Fig. 10 in the revised manuscript) and panels a-c-e (lidar #1) in Fig. 14 (now Fig. 11 in the revised manuscript). But we remove panels b-d-f in Fig. 12 and 13 and put the results of $R^2$ in the plots.

**Specific comments:**

1 L5 "the noise-flattened Doppler spectra." Consider: "the noise-flattened 3-kHz-sampled Doppler spectra".

Thank you for pointing this out. We agree with this comment and incorporated your suggestion in **L8**. The new sentence is "We demonstrate that the rain bias can effectively be removed by normalizing the noise-flattened 3-kHz-sampled Doppler spectra with their peak values before they are averaged down to 50 Hz prior to the determination of the speed.".

**2 L9 Consider: "at 50-Hz (20 ms) temporal resolution"**

Thank you for this suggestion. We agree with this comment and incorporated your suggestion in **L9**. The new sentence is "In comparison to the sonic anemometer measurements acquired at the same location, the wind velocity bias at 50 Hz (20 ms) temporal resolution is reduced from up to $-1.58$ ms$^{-1}$ of the original raw lidar data to $-0.18$ ms$^{-1}$ of the normalized lidar data.".

**3 L10 Please clarify "conventional"**

Thank you for pointing this out. We have changed "conventional" to "original raw" to avoid ambiguity in **L10**. The new sentence is "In comparison to the sonic anemometer measurements acquired at the same location, the wind velocity bias at 50 Hz (20 ms) temporal resolution is reduced from up to $-1.58$ ms$^{-1}$ of the original raw lidar data to $-0.18$ ms$^{-1}$ of the normalized lidar data after suppressing rain signals.".

**4 L25-30 In Sect. Introduction Please comment a bit on the probe-length turbulence averaging effects in comparison to e.g., cup anemometers since this is an important drawback of focusing lidars (e.g., averaging of spatial turbulence scales).**

Thank you for pointing this out. We agree with this comment and incorporated this suggestion in **L25**. The revised sentence is "In-situ cup and sonic anemometers installed on meteorological masts (met masts) can provide only point measurements of wind velocity [Izumi and Barad, 1970]. On the contrary, Doppler lidars can accurately and remotely sense wind velocity by measuring Doppler spectra albeit with their limited ability in measuring turbulence due to probe-length averaging effects [Sathe and Mann, 2013].".

**5 L26 Change "that" into "which"**

Thank you for this suggestion. We agree with this comment. Because the original sentence is quite long, we reformulated the sentence in **L25**. The new sentence is "In-situ cup and sonic anemometers installed on meteorological masts (met masts) can provide only point measurements of wind velocity [Izumi and Barad, 1970]. On the contrary, Doppler lidars can accurately and remotely sense wind velocity by measuring Doppler spectra albeit with their limited ability in measuring turbulence due to probe-length averaging effects [Sathe and Mann, 2013].".

**6 Not sure the acronym "CW" (continuous wave) has previously been defined. Preferably, use "CW" in caps.**

We agree with this. We have defined this abbreviation in **L54** and have incorporated your suggestion in **L72** and figure captions throughout the manuscript. The definition of CW is in the sentence "A field measurement campaign was carried out at Risø where three coherent continuous-wave (CW) Doppler lidars [Mikkelsen et al., 2017] were deployed to point towards a common focus point very close to a mast-mounted sonic anemometer at 31 m height.".

**7 L84 "can be approximated as [REF needed],"**

Thank you for this suggestion. We agree with this comment. The reference is added in **L84** as "The full-width-at-half-maximum (FWHM) of the Lorentzian weighting function or the probe length can be approximated as [Sathe and Mann, 2013],"

$$\text{FWHM} = 2 \cdot z_R = 2 \cdot \frac{\lambda \cdot R^2}{\pi r^2} \tag{1}$$

**8 Tab. 4. Please check if "angle to the North" is computed correctly. In a Cartesian coordinate system, angles are defined positive CCW. And angles between vectors (or between e.g., vector "1" and vector North) are computed by using equipollent vectors so that their origin coincides with the Cartesian origin (i.e., point 5 = projection of point 4 in the XY plane). From the geometrical angles given in Fig.3b and assuming North is 0 deg then I'd say "Angle to North" should be: (WindScanner 1) -42.6 deg; (WindScanner 2) 180 + 7.1.deg = 187.1 deg; (WindScanner 3), 60.7 deg. Please clarify if other Math/Physics conventions are used.**

Thank you for this suggestion. We agree with this comment. However, in the case of our study, we would like to show the lidar's geographic beam direction in Table 1 with the assumption that the North is 0 degrees and clockwise is positive. Therefore, the three lidars' geographic beam directions are 42.6°, 172.9°, and 299.3°. Hope this would be accepted by you.

**9 L95a Please state and clarify to the reader the "key" numbers of the processing. Don't let the reader guess them. Specifically:**

(a) **Fast Fourier Transform (FFT) frequency_resolution: 120 MHz / 512 samples = 234.4 kHz/samples → speed resolution = ($\lambda$/2)*freq_resolution = 0.183 m/s**

(b) **Spectrum_estimation period = sampling rate (1/120 MHz) x 512 (samples/spectrum) x 78 spectrum/average = 332.8e-6 [s] → spectrum_ estimation rate = 1/spectrum_estimation_period = 3 kHz**

Thank you for this suggestion. We agree with this comment and revised **L92 to L101** to emphasize this point. The new sentences are

(a) The backscattered light mixed and amplified by the local oscillator is sampled at a rate of 120 MHz and Doppler spectra containing 512 frequency bins are calculated by Fast Fourier Transform (FFT) with a frequency resolution of $(120\,\text{MHz})/512 = 234.4$ kHz. The wind speed resolution is calculated from this frequency resolution and the laser wavelength $\lambda$, yielding $(1.565\mu\text{m}/2)\cdot(234.4\,\text{kHz}) = 0.183$ ms$^{-1}$.

(b) Subsequently, a block averaging of 78 spectra results in a final sampling period of $512 \cdot 78 \,/(120 \text{ MHz}) = 0.33$ ms, corresponding to a spectrum rate of 3 kHz.

**10 L95b Please briefly summarise which power spectral density (PSD) and which peak spectral estimation method is used to retrieve the Doppler shift. I think this should also be remembered by the reader and shortly discussed later on, in L153-155.**

Thank you for pointing this out. We agree with this comment and have added two sentences, which are "Additionally, Bartlett's method is used to obtain the power spectral density (PSD) of each spectrum [Press et al., 1988, Chap. 13], which is the square of the absolute value of the FFT of the detector's time series. The median method [Held and Mann, 2018] is employed to determine wind velocity." in **L99**. This is mentioned again in **L190** as "the median method is used to determine line-of-sight velocity from the final 50 Hz spectra (Fig. 8c), as it has the least biases for weak signals [Angelou et al., 2012] in comparison to the maximum and centroid methods [Held and Mann, 2018]"

**11 L93 I recommend to repeat "all times are UTC+1" in all figure captions involving time series to help the reader.**

Thank you for pointing this out. We agree with this comment and have added the term "UTC+1" in all figure captions and the text involving time.

**12 L100 "less than the beam transit time of a typical rain drop". Please add literature REFERENCE.**

Thank you for pointing this out. We restructured the sentence to remove "typical". "Typical" here means a large raindrop has the highest falling speed, which is 9 ms$^{-1}$ from the disdrometer measurement in Fig. 6b, not from references. The new sentence in **L102** is "The shortest beam transit time can be determined based on large raindrops' maximum downfall speed of 9 ms$^{-1}$ from the disdrometer measurement in Fig. 6b, the beam width (twice of the beam waist $w_0$), and the elevation angle of a lidar. For lidar #1 with a beam width of 1.12 mm and an elevation angle of 57.9°, the shortest beam transit time is 0.234 ms $= 1.12/(9 \cdot \cos(57.9°))$, while it is 0.362 ms $= 3.14/(9 \cdot \cos(15.3°))$ for lidar #3 with a beam width of 3.14 mm and an elevation angle of 15.3°. Most often, however, raindrops' transit time is longer than the aforementioned shortest time if their paths are away from the lidar focus and if they fall slower. In this study, it is reasonable to set the spectral sampling frequency to 3 kHz so that the sampling period for a spectrum (0.333 ms) is shorter than the beam transit of raindrops [see Jin et al., 2022, Fig. 5b], as shown below. Therefore, the rare instances where a raindrop resides in the beam could be identified and suppressed based on the lidar measurements.".

[Figure]

Figure 2: The geometry of raindrops falling through a focused laser beam. (a) shows raindrops cross the laser beam at random positions, where $w(z)$ is the spot size along the beam, $w_0$ is the beam waist which is 2.35 $mm$ in our case, $z'_1$ and $z'_2$ are two axial distances from the beam's focus, and $z_R$ is the Rayleigh length which is 11.1 $m$. (b) presents the measurements of the power of the back-scattered signal at the Doppler frequency during a raindrop's passage through the beam, following a Gaussian distribution.

**13  L101 Please briefly clarify how the 0.35 ms transit time was estimated. E.g., at 9 m/s fall velocity we get, 9m/s x 3.14 mm = 28 ms (in the near field "waist" of the laser beam).**

You have raised an important point here. However, it should be 3.14mm/(9m/s) to get the transit time. In **L102**, we wrote: "The shortest beam transit time can be determined based on large raindrops' maximum downfall speed of 9 $ms^{-1}$ from the disdrometer measurement in Fig. 6b, the beam width (twice of the beam waist $w_0$), and the elevation angle of a lidar. For lidar #1 with a beam width of 1.12 mm and an elevation angle of 57.9°, the shortest beam transit time is 0.234 ms = $1.12/(9 \cdot \cos(57.9°))$, while it is 0.362 ms = $3.14/(9 \cdot \cos(15.3°))$ for lidar #3 with a beam width of 3.14 mm and an elevation angle of 15.3°. Most often, however, raindrops' transit time is longer than the aforementioned shortest time if their paths are away from the lidar focus and if they fall slower. In this study, it is reasonable to set the spectral sampling frequency to 3 kHz so that the sampling period for a spectrum (0.333 ms) is shorter than the beam transit of raindrops [see Jin et al., 2022, Fig. 5b]. Therefore, the rare instances where a raindrop resides in the beam could be identified and suppressed based on the lidar measurements.".

**14  L109 Consider to give the manufacturer/model for the wind vane and temp sensors.**

Thank you for this suggestion. We agree and have incorporated your suggestion in **L116**. The updated sentence is "Furthermore, the mast is instrumented with a vector wind vane (W200P from Kintech Engineering) at 41 m, and two air temperature sensors (Pt 100, developed by DTU) mounted at 18 m and 70 m, respectively.".

**15** **I recommend drawing X, Y, and Z labels on Fig. 3a to help the reader identify the given unit vectors ($[-0.36, -0.39, -0.85]$, $[-0.10, +0.82, -0.57]$ and $[+0.84, -0.47, -0.26]$).**

Thank you for pointing this out. We agree with this comment and have added X, Y, and Z labels to Fig. 2 now.

**16** **Tab. 2 What does symbol "$/ \geq$" means in "$0.1875/ \geq 8$? Is that a typo?**

Thank you for pointing this out. Yes, it's a typo. We have deleted it in Table 2.

**17** **Fig.7-8. Font size. The "10-min" text in the X and Y labels of panels (b) can barely be seen. Please enlarge the font and include in the captions "Comparison of 10-min wind speed".**

Thank you for this suggestion. We agree and have incorporated your suggestion in the manuscript. Wind speed and direction plots are now merged into one figure (Fig. 5) and we have enlarged the font size. The new figure caption is "Comparison of 10-minute wind measurements with the wind vane, sonic and cup anemometers at several vertical heights. **(a)** 10-minute wind speed by sonic ($SW_{sp}$) and cup ($W_{sp}$) anemometers. **(b)** 10-minute wind direction by sonic anemometers ($S_{dir}$) and the wind vane ($W_{dir}$). **(c)** and **(d)** Linear regression of 10-minute wind speed and direction. The two red lines mark the comparison period of lidar and sonic data from 15:12 to 18:11 (UTC+1).".

**18** **Fig. 9 CAPTIONS (comment to be extended to all manuscript figure captions) I recommend setting label letters at the beginning of each sentence and not at the end or in the middle of the sentence, for better clarity. I recommend to begin each figure caption with a sentence giving an overview of what the figure is about. Then use follow-up labels (a), (b) addressing each panels. E.g., "Rain event September 27th, 2022, 15:00-19:30 measured by the Thies (. . . ). (a) Rain Intensity. (b) Distribution of the number of measurements . . . (color coded)."**
**I also recommend including in the caption the temporal resolution of the data (although this may seem repetitive), e.g., 1 minute, in this case.**

Thank you for pointing this out. We agree with this comment and have changed all the figures' captions in the recommended way and added the temporal resolution.

**19** **L155 Please introduce the acronym "power spectral density (PSD)" See also comment L95b for discussion.**

Thank you for pointing this out. We agree with this comment. We have introduced this in **L101** as "Additionally, Bartlett's method is used to obtain the power spectral density (PSD) of each spectrum [Press et al., 1988, Chap. 13], which is the square of the absolute value of the FFT of the detector's time series.".

**20** **L157-160 Different issues. Unclear. I would suggest expanding the spectral estimation part. In detail:**

(a) "However, if the wind velocity is around zero, this procedure does not work." Why, taking into account that the noise PSD is not zero? Please clarify.

(b) L158-160 "where the line-of-sight velocity fluctuates around zero". At this point in the text, mention to the reader that the vertical line at approx. bin 255 (please clarify bin no.) corresponds to the zero-Doppler shift.

(c) Why does such a negative red peak at bin 255 occur for the background noise (red trace) in Fig. 10a?

Thank you for this suggestion. We agree and have incorporated your suggestion in the manuscript.

(a) We have clarified this sentence: "However, this procedure will not work if the wind velocity is around zero, since the wind Doppler signal would be present on both sides of the zero frequency bin. Then a real, atmospheric Doppler signal would be included in the background spectrum rather than the real background noise." in **L195**.

(b) We have added "(the vertical line at frequency bin 257 corresponding to the zero-Doppler shift in Fig.8)" in **L198**. Now the new sentence is "Therefore, in the case of lidar #1 where the line-of-sight velocity fluctuates around zero (the vertical line at frequency bin 257 corresponding to the zero-Doppler shift in Fig. 8), a background spectrum is calculated for a period where the line-of-sight speed is away from zero.".

(c) It is always 0 value at frequency bin 257 corresponding to zero frequency because the lidar has a high-pass filter that suppresses the near-zero frequency fluctuations. Therefore, there is a negative red peak in Fig. 10a (now Fig. 8 in the revised manuscript). It is an unavoidable feature of continuous-wave lidars.

21 L178 "down sample". Please clarify how the "down-sampling" procedure is carried out. Is it that given the normalized spectra, which make evident the rain returns as very high and narrow peaks, these spectra are screened out for "very large" peaks, therefore, removed from the spectra average? This is known as histogrammed filtering but no quantitative criterion is given. Please give a quantitative criterion for screening out the "rain returns" in Fig. 10d. For example, is that percentile 90 of the cumulative distribution? Please consider to include this histogrammed filtering block in the proposed processing diagram above.

Thank you for pointing this out. We have investigated this method before in our previous work [Jin et al., 2022], by defining an optimal threshold to filter away rain-induced Doppler signals based on the histogram. However, this method is not suitable for all cases since we have to try different thresholds and determine the optimum value when the wind

velocity difference is the smallest compared to sonic data. However, in this paper, we proposed this rain-suppressing normalization method to suppress rain-induced Doppler signals rather than completely sieve them out. Therefore, we changed the term "filter away or filter out" to "suppress" in the manuscript.

**22 Fig. 10. Please vertically align panels (a) and (c). Please use the same X-axis range to ease comparison.**

Thank you for this suggestion. We agree and have incorporated your suggestion in the manuscript. Now panels (a) and (c) are aligned in Fig. 10 (now Fig. 8 in the revised manuscript) and the same frequency range for panels (a), (b), and (c) is used.

**23 Fig. 10 caption. Please add: "The solid black line stands for the zero frequency bin" (as in Fig. 11).**

Thank you for this suggestion. We agree and have incorporated your suggestion in the manuscript. The new caption is "Examples of representative Doppler spectra measured at the moderate-rain minute (15:48, UTC+1) with the highest rain intensity. **(a)** A 3-kHz-sampled spectrum containing only wind signal (blue) and the mean background spectrum (red). **(b)** A 3-kHz-sampled spectrum containing rain signal (blue) and the mean background spectrum (red). **(c)** A noise-flattened 50-Hz-sampled spectrum and its spectral threshold. **(d)** Histogram of the maximum spectral energy $S_{max}$ of 180000 raw spectra over the duration of the same minute with a red circle marking the strongest rain signals. The solid black line stands for the zero-Doppler shift at frequency bin 257." (same as Fig. 9 in the revised manuscript).

**24 Fig. 10. Which method is used to compute the PSD? E.g. Periodogram or others. Please include literature reference.**

Thank you for pointing this out. It is Bartlett's method to compute the PSD. We added the information in **L99** as "Additionally, Bartlett's method is used to obtain the power spectral density (PSD) of each spectrum [Press et al., 1988, Chap. 13], which is the square of the absolute value of the FFT of the detector's time series.".

**25 Fig. 11 Caption, L4. "represent the median frequency bin" or "stand for the median frequency bin"**

Thank you for this suggestion. We agree. The new caption is "The red and blue dashed lines represent the median frequency bin of the raw and the normalized Doppler spectra, which are used to derive line-of-sight wind velocity.".

**26 Fig. 11 (b)(d) Please use the same frequency-range (X-axis) in both panels.**

Thank you for this suggestion. We agree. We have used the same frequency range for panels (b) and (d) in Fig. 11 (now Fig. 9 in the revised manuscript).

**27 L209 Change "that occurred" into "which occurred"**

Thank you for this suggestion. We agree. However, we have deleted this sentence because another reviewer suggested that such information should be added in the section

that introduces sonic anemometers. Therefore, we removed this part to **L127** in Section 2.2 as "It is evident from the sonic status information that wind velocity measurements by sonic anemometers can be affected by raindrops. In those cases, the sonic anemometer would repeat the previous velocity value and the status would be "4". Thus, the linear interpolation method was used in this study to eliminate repeated velocities, which represented about 60% of the 50 Hz sonic data recorded at moderate-rain minutes.".

28 **L222 "more" missing → it rains more heavily than lightly**
Thank you for pointing this out. We agree. However, L241 was written as "it rains more heavily than lightly", but we changed "more" to "it rains more heavily than lightly" in **L280**. The new sentence is "These lead to the same conclusions discussed previously that rain-suppressing normalization performs well for the large probe length when it rains as well as for the small probe length when it rains more heavily than lightly.".

29 **Tabs. 4-5 Please repeat in the caption key information on "rain intensity" and "probe length".**
Thank you for this suggestion. We agree and have added the probe length and rain intensity in Tables 4 and 5. The new caption of Table 4 is "1-minute averaged wind velocity based on 50 Hz data and the corresponding bias between the sonic anemometer and WindScanner lidar #3 (probe length of 9.8 m) at three minutes, with (norm) and without (raw) normalization. Rain intensity at the light-rain and moderate-rain minutes are 1 mmh$^{-1}$ and 4 mmh$^{-1}$.", same as Table 5.

30 **Tab. 4. Please stick to two decimal digits everywhere. Please check for errors/typos in column "light-rain minute"**
Thank you for pointing this out. We agree and have changed all the numbers with two decimal digits and corrected the calculation errors from 0.11 and 0.14 to 0.01 in the third row in Table 4.

**Table 4.** 1-minute averaged wind velocity based on 50 Hz data and the corresponding bias between the sonic anemometer and lidar #1 (probe length of 1.2 m) at three minutes, with (norm) and without (raw) normalization. Rain intensity at the light-rain and moderate-rain minutes are 1 mmh$^{-1}$ and 4 mmh$^{-1}$.

| | $V_{sonic}$ (ms$^{-1}$) | $V_{raw}$ (ms$^{-1}$) | $V_{sonic} - V_{raw}$ (ms$^{-1}$) | $V_{norm}$ (ms$^{-1}$) | $V_{sonic} - V_{norm}$ (ms$^{-1}$) |
|---|---|---|---|---|---|
| No-rain minute 15:13:20+1min | -1.01 | -1.07 | 0.06 | -1.08 | 0.07 |
| Light-rain minute 16:36:20+1min | -0.38 | -0.39 | 0.01 | -0.39 | 0.01 |
| Moderate-rain minute 15:48:20+1min | -0.64 | -0.49 | -0.15 | -0.60 | -0.04 |

31 **Figs. 12-13. Legends: The logical order should be "sonic-raw-norm" instead of "raw-sonic-norm".**
Thank you for this suggestion. We agree with this point and have incorporated your

suggestion in the manuscript. We have changed the legend order to "sonic-raw-norm" in Fig. 12 (now Fig.10) and 13 (now Fig.11).

**32 Fig. 16 caption. Please repeat in the caption "probe length" values to help the reader.**

Thank you for this suggestion. We agree and have added the probe length in the caption, which is now "Difference of 1-minute averaged wind velocity between lidar and sonic measurements together with the rain intensity (the solid black curve) from 15:12 to 18:11 (UTC+1). **(a)** Lidar #1 with the probe length of 1.2 m. **(b)** Lidar #3 with the probe length of 9.8 m. The raw and normalized lidar data are marked in red and blue.".

**33 CONCLUSIONS Please give conclusions on your findings about the performance of the methods for different probe lengths (short/large probe length, which is an important point –although more risky- or your research) as well as future lines. Part of the conclusions given in L183-185, L221-224, L244 should be rewritten / summarised in Sect. Conclusions.**

You have raised an important point here. We agree and have added the findings regarding the different probe lengths, the conclusions from L183-185, L221-224, and L244 as well as some outlooks. Therefore, the new CONCLUSIONS is "In this paper, we have shown an experimental proof-of-concept demonstration of a method to reduce the bias caused by precipitation on continuous-wave Doppler lidar measurements of wind speed. This is accomplished by sampling Doppler spectra faster than most raindrops' beam transit time, which in the current case was at 3 kHz. Subsequently, the 3 kHz spectra are normalized with their peak values to suppress strong backscatter signals from raindrops before being averaged down to 50 Hz from which the radial wind velocity is determined.

Results from lidar beams with different elevation angles and focus distances were studied under different rain intensities measured by a disdrometer. The derived wind velocities were compared with a sonic anemometer reference. From the comparison, we find that the rain-suppressing normalization has the most significant impact on reducing bias when the probe volume (growing with the fourth power of the focus distance) is the largest. However, when the probe volume is small (shorter focus distances), the impact of rain is limited. Rain-induced bias also varies according to elevation angle but to a lesser extent. However, the exact nature of these relations remains to be further verified and understood. The tendency is that the more it rains, the stronger the bias and the more the rain-suppressing normalization is reducing the bias. For moderate rain intensity (we do not have a heavy rain period in our data), the range of the bias is reduced from the interval 0.1 to 0.4 ms$^{-1}$ to 0.0 to 0.1 ms$^{-1}$. The suggested method in this study could also be investigated for rain events (containing heavy rain) on several days and also for pulsed Doppler lidars even though their measurement volume is quite larger than that of the continuous-wave lidars. Further investigations could also attempt to retrieve the falling velocity and the size distribution of raindrops using the fast Doppler spectra.".

**Bibliography**

N. Angelou, F. F. Abari, J. Mann, T. Mikkelsen, and M. Sjöholm. Challenges in noise removal from doppler spectra acquired by a continuous-wave lidar. In *Proceedings of the 26th International Laser Radar Conference, Porto Heli, Greece*, pages 25–29, 2012.

S. Davoust, A. Jehu, M. Bouillet, M. Bardon, B. Vercherin, A. Scholbrock, P. Fleming, and A. Wright. Assessment and optimization of lidar measurement availability for wind turbine control. Technical report, National Renewable Energy Lab.(NREL), Golden, CO (United States), 2014.

M. Debnath, G. V. Iungo, R. Ashton, W. A. Brewer, A. Choukulkar, R. Delgado, J. K. Lundquist, W. J. Shaw, J. M. Wilczak, and D. Wolfe. Vertical profiles of the 3-d wind velocity retrieved from multiple wind lidars performing triple range-height-indicator scans. *Atmospheric Measurement Techniques*, 10(2):431–444, 2017.

Glossary of Meteorology (June 2000). Rain. https://glossary.ametsoc.org/wiki/Rain, last access: 21 June 2023.

F. Guo, J. Mann, A. Peña, D. Schlipf, and P. W. Cheng. The space-time structure of turbulence for lidar-assisted wind turbine control. *Renewable Energy*, 2022.

D. P. Held and J. Mann. Comparison of methods to derive radial wind speed from a continuous-wave coherent lidar doppler spectrum. *Atmospheric Measurement Techniques*, 11(11):6339–6350, 2018.

Y. Izumi and M. L. Barad. Wind speeds as measured by cup and sonic anemometers and influenced by tower structure. *Journal of Applied Meteorology and Climatology*, 9(6):851–856, 1970.

D. Jena and S. Rajendran. A review of estimation of effective wind speed based control of wind turbines. *Renewable and Sustainable Energy Reviews*, 43:1046–1062, 2015.

L. Jin, N. Angelou, J. Mann, and G. C. Larsen. Improved wind speed estimation and rain quantification with continuous-wave wind lidar. In *Journal of Physics: Conference Series*, volume 2265, page 022093. IOP Publishing, 2022.

J. Li, X. Wang, and X. B. Yu. Use of spatio-temporal calibrated wind shear model to improve accuracy of wind resource assessment. *Applied energy*, 213:469–485, 2018.

T. Mikkelsen, M. Sjöholm, N. Angelou, and J. Mann. 3d windscanner lidar measurements of wind and turbulence around wind turbines, buildings and bridges. In *IOP Conference Series: Materials Science and Engineering*, volume 276, page 012004. IOP Publishing, 2017.

W. H. Press, W. T. Vetterling, S. A. Teukolsky, and B. P. Flannery. *Numerical recipes*. Citeseer, 1988.

S. Samadianfard, S. Hashemi, K. Kargar, M. Izadyar, A. Mostafaeipour, A. Mosavi, N. Nabipour, and S. Shamshirband. Wind speed prediction using a hybrid model of the multi-layer perceptron and whale optimization algorithm. *Energy Reports*, 6:1147–1159, 2020.

A. Sathe and J. Mann. A review of turbulence measurements using ground-based wind lidars. *Atmospheric Measurement Techniques*, 6(11):3147–3167, 2013.

K. Träumner, J. Handwerker, A. Wieser, and J. Grenzhäuser. A synergy approach to estimate properties of raindrop size distributions using a doppler lidar and cloud radar. *Journal of Atmospheric and Oceanic Technology*, 27(6):1095–1100, 2010.

M. Türk and S. Emeis. The dependence of offshore turbulence intensity on wind speed. *Journal of Wind Engineering and Industrial Aerodynamics*, 98(8-9):466–471, 2010.

A. Van Ulden and A. Holtslag. Estimation of atmospheric boundary layer parameters for diffusion applications. *Journal of Applied Meteorology and Climatology*, 24(11):1196–1207, 1985.

T. Wei, H. Xia, J. Hu, C. Wang, M. Shangguan, L. Wang, M. Jia, and X. Dou. Simultaneous wind and rainfall detection by power spectrum analysis using a vad scanning coherent doppler lidar. *Optics express*, 27(22):31235–31245, 2019.

---

## Author Comment (AC4)

**0.1 Response to Reviewer 3 Comments**

**Dear reviewer:**

We appreciate the time and effort that you have dedicated to providing your insightful comments on our paper. We have been able to incorporate changes to reflect most of the suggestions provided by you. We have highlighted the changes within the manuscript.

**Specific comments:**

1 **Line 5 and Line 139: If this paper could provide more cases or results on several days with various rain intensities (containing light rain, moderate rain, and heavy rain) would make the conclusion more convincing.**
Thank you for pointing this out. We agree with this comment and we would like to investigate more on several days with various rain intensities in this proof-of-concept study. Unfortunately, we don't have heavy-rain data. However, this is the first study to sample the Doppler spectrum very fast up to 3 kHz and normalize each spectrum by its peak value to suppress Doppler signals generated by raindrops. Before this study, we have already conducted another field measurement with one continuous-wave lidar for quite a short period [Jin et al., 2022]. In this study, we compared three-hour data and the results are promising. We added some outlooks in the Conclusion part "The tendency is that the more it rains, the stronger the bias and the more the rain-suppressing normalization is reducing the bias. For moderate rain intensity (we do not have a heavy rain period in our data), the range of the bias is reduced from the interval 0.1 to 0.4 ms$^{-1}$ to 0.0 to 0.1 ms$^{-1}$. The suggested method in this study could also be investigated for rain events (containing heavy rain) on several days and also for pulsed Doppler lidars even though their measurement volume is quite larger than that of the continuous-wave lidars. Further investigations could also attempt to retrieve the falling velocity and the size distribution of raindrops using the fast Doppler spectra.".

2 **Line 100: How do the 0.35 ms of the raindrops' beam transit time calculate? Please clarify.**
Thank you for pointing this out. In **L102**, we wrote: "The shortest beam transit time can be determined based on large raindrops' maximum downfall speed of 9 ms$^{-1}$ from the disdrometer measurement in Fig. 6b, the beam width (twice of the beam waist $w_0$), and the elevation angle of a lidar. For lidar #1 with a beam width of 1.12 mm and an elevation angle of 57.9°, the shortest beam transit time is 0.234 ms = $1.12/(9 \cdot \cos(57.9°))$, while it is 0.362 ms = $3.14/(9 \cdot \cos(15.3°))$ for lidar #3 with a beam width of 3.14 mm and an elevation angle of 15.3°. Most often, however, raindrops' transit time is longer than the aforementioned shortest time if their paths are away from the lidar focus and if they fall slower. In this study, it is reasonable to set the spectral sampling frequency to 3 kHz so that the sampling period for a spectrum (0.333 ms) is shorter than the beam transit of

raindrops [see Jin et al., 2022, Fig. 5b]. Therefore, the rare instances where a raindrop resides in the beam could be identified and suppressed based on the lidar measurements.".

3 **Line 158: "... where the line-of-sight speed is away from zero." Please clarify and explain the reason for this processing.**
Thank you for this comment. We agree with this comment and explained it in **L195**. The explanation is "However, this procedure will not work if the wind velocity is around zero, since the wind Doppler signal would be present on both sides of the zero frequency bin. Then a real, atmospheric Doppler signal would be included in the background spectrum rather than the real background noise. Therefore, in the case of lidar #1 where the line-of-sight velocity fluctuates around zero (the vertical line at frequency bin 257 corresponding to the zero-Doppler shift in Fig. 8), a background spectrum is calculated for a period where the line-of-sight speed is away from zero.".

4 **Line 169: This paper mentioned the rain spectrum with a high value of PSD and a narrow peak. However, considering the strong attenuation of laser energy caused by the raindrops, sometimes the PSD of rain spectra gets weak and has the nearly same magnitude as the aerosol spectrum. How to distinguish the wind and rain in these cases?**
You have raised a good point. That is the limitation of our method because we could not distinguish the two signals with similar magnitude. Therefore, there is still a bias between lidar data and sonic data after applying this rain-suppressing method. But the bias is reduced. The research objective of this study is to reduce the adverse impact of raindrops when measuring wind velocities by normalizing the fast Doppler spectra with their peak values.

5 **Is this method proposed in this paper also suitable for pulsed Doppler lidar?**
Thank you for this comment. It could be investigated with pulsed Doppler lidar even though this would be difficult. We added several sentences in the Conclusion part as "The suggested method in this study could also be investigated for rain events (containing heavy rain) on several days and also for pulsed Doppler lidars even though their measurement volume is quite larger than that of the continuous-wave lidars. Further investigations could also attempt to retrieve the falling velocity and the size distribution of raindrops using the fast Doppler spectra.".

6 **This paper evaluates the performance of this method under several rain intensities. How about the influence of horizontal velocity on the results? Because a big raindrop will break up more small raindrops with high wind speed.**
Thank you for this comment. You have raised a good point. We would like to evaluate the influence of horizontal velocity on the retrieved wind velocities. However, in our study, we investigated a method to reduce the influence of raindrops in wind velocity measurements by CW lidars and how this proposed rain-suppressing normalization method performs in

reducing the bias compared to sonic data. Further experiments may be able to shed light on this issue.

**Bibliography**

L. Jin, N. Angelou, J. Mann, and G. C. Larsen. Improved wind speed estimation and rain quantification with continuous-wave wind lidar. In *Journal of Physics: Conference Series*, volume 2265, page 022093. IOP Publishing, 2022.

---

## Author Comment (AC5)

**0.1 Response to Reviewer 4 Comments**

**Dear reviewer:**

We appreciate the time and effort that you have dedicated to providing your insightful comments on our paper. We have been able to incorporate changes to reflect most of the suggestions provided by you. We have highlighted the changes within the manuscript.

**Major comments:**

1 **The main critical point of this manuscript is that the conclusion was based only on the values of three minutes with (no) rain during one event. However, more minutes with rain are existing (Figure 16), but only a visual comparison was provided. It is essential to validate the proposed data processing procedure by considering more rain minutes (if possible also from other rain events on different dates with different intensities) to conclude the applicability of the method in various rain conditions.**

Thank you for pointing this out. In this study, we first compared three minutes of data with different rain intensities in Section 5.1 and then applied the rain-suppressing normalization method to three-hour data in Section 5.2. We would like to investigate more data on several days with various rain intensities for this proof-of-concept study. This point is presented in the Conclusion part as "The tendency is that the more it rains, the stronger the bias and the more the rain-suppressing normalization is reducing the bias. For moderate rain intensity (we do not have a heavy rain period in our data), the range of the bias is reduced from the interval 0.1 to 0.4 ms$^{-1}$ to 0.0 to 0.1 ms$^{-1}$. The suggested method in this study could also be investigated for rain events (containing heavy rain) on several days and also for pulsed Doppler lidars even though their measurement volume is quite larger than that of the continuous-wave lidars. Further investigations could also attempt to retrieve the falling velocity and the size distribution of raindrops using the fast Doppler spectra.".

2 **From the text, it is not easy to distinguish between the steps in the standard procedure of WindScanner data processing and the new proposed procedure. As this is the key point in this manuscript, a sketch showing the steps with and without rain-signal exclusion would be really helpful to understand the differences in the data processing.**

Thank you for pointing this out. We agree with this comment. We added a processing diagram block in the draft to show the spectral process steps of our proposed method. Now, Sect. 4.1 is restructured according to the suggestion. The first paragraph is about how Doppler spectra after being averaged to lower frequencies are processed. Then we show the comparison between normal Doppler spectra with only aerosol-induced Doppler

signals and the spectra with rain-induced signals. Subsequently, we proposed our rain-suppressing normalization method.

[Figure]

Figure 1: Processing block diagram of the rain-suppressing normalization method (the solid lines from ① to ③) to estimate wind velocity based on 3-kHz-sampled Doppler spectra. Doppler spectra at lower frequencies that do not resolve individual raindrops (like 50 Hz) are processed according to the purple path including the dashed purple line, ②, and ③.

3 **The authors provide 0.35 ms as transit time of a raindrop through the lidar beam. Which assumptions were made to calculate this time? It would be interesting to have a range of potential transit times, because different raindrop sizes exist. Depending on the size and other factors, the fall velocity of a raindrop varies as shown already in Figure 9 (b).**
Thank you for pointing this out. We agree with this comment and explained in the text like this "The shortest beam transit time can be determined based on large raindrops' maximum downfall speed of 9 ms$^{-1}$ from the disdrometer measurement in Fig. 6b, the beam width (twice of the beam waist $w_0$), and the elevation angle of a lidar. For lidar #1 with a beam width of 1.12 mm and an elevation angle of 57.9°, the shortest beam transit time is 0.234 ms = $1.12/(9 \cdot \cos(57.9°))$, while it is 0.362 ms = $3.14/(9 \cdot \cos(15.3°))$ for lidar #3 with a beam width of 3.14 mm and an elevation angle of 15.3°. Most often, however, raindrops' transit time is longer than the aforementioned shortest time if their paths are away from the lidar focus and if they fall slower. In this study, it is reasonable to set the spectral sampling frequency to 3 kHz so that the sampling period for a spectrum (0.333 ms) is shorter than the beam transit of raindrops [see Jin et al., 2022, Fig. 5b]. Therefore, the rare instances where a raindrop resides in the beam could be identified and suppressed based on the lidar measurements.".

4 **The PDFs of the no-rain minutes were higher in case of the new procedure compared to the old one. Can the authors elaborate on the reasons for that and possible consequences? This example shows that a validation with more data is necessary to see how the data processing procedure behaves in non-rainy periods as well. This is important, because it raises the question if the proposed procedure can only be applied for measurements taken during rain or if it can be applied during dry and wet conditions.**

Thank you for this comment. Due to the fact that this minute was a few minutes before rain began, it is possible that a raindrop passed through the laser beam of lidar #1 and was detected by the lidar. After applying the proposed method, the strong rain signal was suppressed. This results in a higher peak in the blue curve than in the red, but closer to the green (the sonic data). From the results in Tables 4 and 5 as well as Fig. 10, we believe that our proposed method also works for dry conditions as the bias between the sonic anemometer and two lidars' measurements is almost the same.

**Minor comments and technical corrections:**

1 **L10: The authors write significant reduction. Was the reduction analyzed with a statistical test that supports the assumption of a significant reduction? If yes, please include this result in the manuscript. If not, please consider removing the word 'significant'.**
Thank you for pointing this out. We agree with this comment and removed the word "significant" in **L11**. Now the sentence is "This reduction of the bias occurs at the minute with the highest amount of rain when the measurement distance of the lidar is 103.9 m with a corresponding probe length being 9.8 m.".

2 **L11: It is not clear what should be understood by 'the measurement distance of the lidar'. Distance to what?**
Thank you for this comment. Here the measurement distance is the focus distance of a lidar. We have changed "measurement" to "focus" in **L12**. Now the new sentence is "This reduction of the bias occurs at the minute with the highest amount of rain when the focus distance of the lidar is 103.9 m with a corresponding probe length being 9.8 m.".

3 **L16-22: When starting with meteorology, examples of this application area should be mentioned first.**
Thank you for this comment. We agree and have implemented it in the manuscript. The new paragraph is "Precise determination of wind flow plays an important role in reducing loads on critical turbine components and power variations, correcting commonly used models for wind energy assessment, improving the performance of wind turbine controllers, and improving the prediction of the potential wind power extracted from the wind [Davoust et al., 2014, Jena and Rajendran, 2015, Li et al., 2018, Samadian-fard et al., 2020, Guo et al., 2022]. Besides, wind velocity estimation is also useful for understanding important phenomena, i.e., atmospheric boundary layer flows and wind turbulence [Van Ulden and Holtslag, 1985, Türk and Emeis, 2010, Debnath et al., 2017]. Therefore, accurate measurements of wind velocity are crucial for many applications in meteorology and wind energy.".

4 **L42: I assume the measurements of Doppler lidars are influenced, not the instrument itself? Maybe the authors can clarify that.**

Thank you for this comment. We agree and have implemented this in the manuscript in **L42**. The new sentence is "Doppler lidars' measurements of wind velocity can be influenced by heavy rainfall because the projected speed of raindrops on the propagation direction of the lidar beam will be different from the line-of-sight wind velocity.".

5 **L43-L44: Please remove the brackets around the reference of Träumner et al.**
Thank you for pointing this out. We agree and removed the brackets around the reference to Träumner et al. The new sentence is "A synergy approach was proposed by Träumner et al. [2010], which combined radar and vertically scanning lidar measurements to estimate the vertical wind velocity and the raindrop size distribution during rain episodes.".

6 **L46: Please remove the brackets around the reference Wei et al.**
Thank you for this comment. We agree and removed the brackets. The new sentence is "Later, by using a velocity-azimuth display (VAD) scanning technique, wind speed, and rainfall speed were simultaneously retrieved in Wei et al. [2019], by fitting the two-peak spectrum with a two-component Gaussian model. The spectral peak close to 0 ms$^{-1}$ is the Doppler signal of the vertical wind speed, which can be easily recognized in this scenario.".

7 **L54: The acronym 'cw' is not defined. Please add the information.**
Thank you for this comment. We agree with this point. The definition of CW is in **L54** "A field measurement campaign was carried out at Risø where three coherent continuous-wave (CW) Doppler lidars [Mikkelsen et al., 2017] were deployed to point towards a common focus point very close to a mast-mounted sonic anemometer at 31 m height.".

8 **L107: What does Risø in the brackets mean? Is this the type/manufacturer of the cup anemometers? Please clarify.**
Thank you for this comment. We have added the type information in **L113**. The new sentence is "There are five sonic anemometers (USA-1, Metek) on booms facing north and five cup anemometers (P2546A from WindSensor) on booms facing south, placed at 18 m, 31 m, 44 m, 57 m, and 70 m above the terrain (Fig. 3). The sampling frequency of the sonic anemometers was 50 Hz.".

9 **L108-L109: What is the manufacturer of the wind vane and the air temperature sensor? In this connection, the wording 'absolute temperature' sounds strange. Maybe 'air temperature' is more appropriate?**
Thank you for this comment. We have added the type information in **L116** and deleted the word "absolute". The new sentence is "Furthermore, the mast is instrumented with a vector wind vane (W200P from Kintech Engineering) at 41 m, and two air temperature sensors (Pt 100, developed by DTU) mounted at 18 m and 70 m, respectively.".

10 **L138-L139: It is not clear how the wake influence of the turbine was determined and why this was important for the experiment. Please clarify.**

Thank you for this comment. So far, we are uncertain about the influence of turbine wake on our proposed rain-suppressing normalization method. But, we would like to have clean data to investigate the performance of the suggested method. Therefore, we avoided the complication of turbine wakes.

**11 L144: Can the authors provide a reference to the Met Office's definition?**

Thank you for this comment. We agree with this point and add a reference to the definition in **L172**. The new sentence is "Moderate rain is defined as a precipitation rate between 2.6 mm and 7.6 mm per hour [Glossary of Meteorology (June 2000), last access: 21 June 2023.].".

**12 L185: Please remove the brackets around the reference of Angelou et al.**

Thank you for this comment. e agree and removed the brackets. The new sentence is "As concluded in Angelou et al. [2012], the optimum number of standard deviations for defining the threshold is not the same for different data sets and a number of 2.5 has been used for the three lidars in this investigation.".

**13 L186: Why was the number 2.5 used for the analysis?**

Thank you for this comment. We compared the velocity difference between sonic data and lidar data with different values ranging from 1.0 to 4.5. We find that 2.5 is a reasonable number for all three lidars. We added some explanation to the manuscript "After calculating velocity difference with sonic data over a short period of time, a number of 2.5 has been chosen for the three lidars in this study." in **L205**.

[Figure]

Figure 2: Mean absolute 1-minute wind velocity difference between the three lidars and the sonic anemometer as a function of different multiples of the standard deviation.

**14 L195-L197: It is not clear what the authors want to express with the sentence starting with 'Consequently, the projection of . . . '. Maybe a sketch could help?**

Thank you for this comment. The sketch is as follows. Because another reviewer has pointed out that there are many figures in the manuscript, we explained this point in the text instead of putting a figure: "It is worth noting that the wind direction at the minute with the highest rain intensity (15:48, UTC+1) is from 160° by the 10-minute averaged sonic data, and the two lidars' geographic beam directions are 42.6° and 299.3° (Fig. 2). Therefore, the wind is moving away from both lidars' laser beams at this minute, causing negative line-of-sight velocity. Consequently, the projection of the resultant velocity of raindrops, in the measuring configuration used here, is smaller than that of the horizontal wind speed in the beam direction.". Hope this will be accepted by you.

[Figure]

Figure 3: Illustration of velocity projection of aerosol speed $Vel_{aero}$ and resultant raindrop speed $Vel_{rain\_resul}$ on the propagation direction of one lidar.

**15 L204: '... in detail' instead of 'in details'.**

Thank you for this comment. We agree and have implemented this in the manuscript in **L222**. The new sentence is "In the section below, we compare the radial wind velocity detected by lidars and the sonic anemometer at 31 m height in detail in light of the promising results about the effective suppression of rain Doppler signals at one moderate-rain minute (15:48, UTC+1).".

**16 L208-L210: How much do raindrops influence the sonic measurements? The authors should provide some information about that in the sensor description in Chapter 2. Furthermore, what interpolation method was used?**

Thank you for this comment. If the sonic measurement is influenced by raindrops, its status will be "4", indicating that this is not a valid number, and it just repeats the previous number. Therefore, we used linear interpolation to replace the repeated numbers. We added one paragraph in **L127** in Section 2.2 as "It is evident from the sonic status information that wind velocity measurements by sonic anemometers can be affected by raindrops. In those cases, the sonic anemometer would repeat the previous velocity value and the status would be "4". Thus, the linear interpolation method was used in this study to eliminate repeated velocities, which represented about 60% of the 50 Hz sonic data recorded at moderate-rain minutes.".

**17 Figures in general: It would be easier to read the caption if (a), (b), ... are written before the actual description.**

Thank you for this comment. We agree and have changed all figures' captions to have (a), (b), ... written before the description.

**18 Figure 1: Can the authors add the information about the location of the disdrometer?**

Thank you for pointing this out. We have removed Fig. 1 from the manuscript. But we added the location information in **L138** as "This disdrometer was about 20 m north of the met mast."

**19 Figure 2: Do the red arrows indicate the location of the common focus point on the met mast? Please add some explanation about the arrows in the figure caption.**

Thank you for this comment. Yes, they indicated the common focus point. We have written "Blue points marked by 1, 2, and 3 are the three CW Doppler lidars, focused at the common point 4 which is 1 m north of the sonic anemometer at a height of 31 m above the ground." in the caption.

**20 Figure 6: The disdrometer shown on this photo is not a Thies LPM, but a Ott Parsivel2. Please check the manufacturer of the disdrometer which was used in this experiment.**

You have raised an important point here. Yes, we put the wrong picture before. It is corrected in the manuscript now.

[Figure]

Figure 4: Thies Laser Precipitation Monitor(LPM) at DTU Risø campus.

**21 Figure 9: Is the plotted rain intensity taken from the automatic output of the disdrometer or calculated based on a quality-controlled rain-drop-size distribution?**

Thank you for this comment. The plotted rain intensity in Fig. 9 (now Fig. 6) is taken from the automatic output of the disdrometer.

**22 Figure 10: It is a bit confusing using the same colours in (a), (b) and (c), although the colours in (c) describe not the same as in (a) and (b). The authors should consider using other colours or adding a legend to (c). Furthermore, the acronym 'PSD' is not described. This information should be added.**

Thank you for this comment. We agree and have changed the color in panel (c). The acronym 'PSD' is defined in **L99** as Additionally, Bartlett's method is used to obtain the power spectral density (PSD) of each spectrum [Press et al., 1988, Chap. 13], which is the square of the absolute value of the FFT of the detector's time series..

[Figure]

Figure 5: Examples of representative Doppler spectra measured at the moderate-rain minute (15:48, UTC+1) with the highest rain intensity. **(a)** A 3-kHz-sampled spectrum containing only wind signal (blue) and the mean background spectrum (red). **(b)** A 3-kHz-sampled spectrum containing rain signal (blue) and the mean background spectrum (red). **(c)** A noise-flattened 50-Hz-sampled spectrum and its spectral threshold. **(d)** Histogram of the maximum spectral energy $S_{max}$ of 180000 raw spectra over the duration of the same minute with a red circle marking the strongest rain signals. The solid black line stands for the zero-Doppler shift at frequency bin 257.

**23 Figure 11: Strictly speaking, the Doppler signal is caused by aerosols not by wind.**
Thank you for pointing this out. We agree and have changed to "Doppler signals by aerosols" in Fig. 8 (the above figure).

**24 Figure 12 and Figure 13: To the last sentence the information 'in the scatter plot' should be added to make the description clearer.**
Thank you for this comment. We agree. However, we have removed the scatter plots in Fig. 12 and 13.

**25** **Figure 15:** **The figure could be simplified by plotting the bars in the same direction and the two different methods (SonicToRaw and SonicToNorm) are visualised by different colours (e.g. bright and dark). This would allow an easier comparison of the values.**
Thank you for this comment. We agree and the new figure is as follows.

[Figure]

(a)                                                                      (b)

Figure 6: Comparison of the integral value of the PDF's absolute difference between the sonic and the lidar data with (SonicToNorm) and without (SonicToRaw) rain-suppressing normalization at no-rain, light-rain ($I_{rain} = 1$ mmh$^{-1}$), and moderate-rain ($I_{rain} = 4$ mmh$^{-1}$) minutes. **(a)** Lidar #1. **(b)** Lidar #3.

**26** **Table 4:** **For 'Light-rain minute 16:36' in two cases three digits are given. Depending on the possible accuracy, please provide two or three digits for all numbers.**
Thank you for pointing this out. We agree and have implemented this in Table 4.

**Table 4.** 1-minute averaged wind velocity based on 50 Hz data and the corresponding bias between the sonic anemometer and lidar #1 (probe length of 1.2 m) at three minutes, with (norm) and without (raw) normalization. Rain intensity at the light-rain and moderate-rain minutes are 1 mmh$^{-1}$ and 4 mmh$^{-1}$.

|  | $V_{sonic}$ (ms$^{-1}$) | $V_{raw}$ (ms$^{-1}$) | $V_{sonic} - V_{raw}$ (ms$^{-1}$) | $V_{norm}$ (ms$^{-1}$) | $V_{sonic} - V_{norm}$ (ms$^{-1}$) |
|---|---|---|---|---|---|
| No-rain minute 15:13:20+1min | -1.01 | -1.07 | 0.06 | -1.08 | 0.07 |
| Light-rain minute 16:36:20+1min | -0.38 | -0.39 | 0.01 | -0.39 | 0.01 |
| Moderate-rain minute 15:48:20+1min | -0.64 | -0.49 | -0.15 | -0.60 | -0.04 |

**27** **Table 4 & 5:** **Are the values calculated for the same time period plotted in Figure 12 and Figure 13? The figures represent the values for a bit more than**

exact one minute. The authors are asked to state exactly which time period (including seconds) was used for the values provided in the tables.

Thank you for this comment. We agree with this point. Both figures and tables are compared with the same one-minute period, for example, 15:13:20+1min for the no-rain minute. Now the tables are corrected to be consistent with the figures.

**Table 4.** 1-minute averaged wind velocity based on 50 Hz data and the corresponding bias between the sonic anemometer and lidar #1 (probe length of 1.2 m) at three minutes, with (norm) and without (raw) normalization. Rain intensity at the light-rain and moderate-rain minutes are 1 mmh$^{-1}$ and 4 mmh$^{-1}$.

| | $V_{sonic}$ (ms$^{-1}$) | $V_{raw}$ (ms$^{-1}$) | $V_{sonic} - V_{raw}$ (ms$^{-1}$) | $V_{norm}$ (ms$^{-1}$) | $V_{sonic} - V_{norm}$ (ms$^{-1}$) |
|---|---|---|---|---|---|
| No-rain minute 15:13:20+1min | -1.01 | -1.07 | 0.06 | -1.08 | 0.07 |
| Light-rain minute 16:36:20+1min | -0.38 | -0.39 | 0.01 | -0.39 | 0.01 |
| Moderate-rain minute 15:48:20+1min | -0.64 | -0.49 | -0.15 | -0.60 | -0.04 |

**Table 5.** 1-minute averaged wind velocity based on 50 Hz data and the corresponding bias between the sonic anemometer and lidar #3 (probe length of 9.8 m) at three minutes, with (norm) and without (raw) normalization. Rain intensity at the light-rain and moderate-rain minutes are 1 mmh$^{-1}$ and 4 mmh$^{-1}$.

| | $V_{sonic}$ (ms$^{-1}$) | $V_{raw}$ (ms$^{-1}$) | $V_{sonic} - V_{raw}$ (ms$^{-1}$) | $V_{norm}$ (ms$^{-1}$) | $V_{sonic} - V_{norm}$ (ms$^{-1}$) |
|---|---|---|---|---|---|
| No-rain minute 15:13:20+1min | -5.42 | -5.41 | -0.01 | -5.45 | 0.03 |
| Light-rain minute 16:36:20+1min | -3.37 | -3.16 | -0.21 | -3.36 | -0.01 |
| Moderate-rain minute 15:48:20+1min | -3.62 | -3.29 | -0.33 | -3.54 | -0.08 |

28 The authors are not consistent by using 'rain drops' and 'rain droplets'. Please harmonize the description to 'rain drops'.

Thank you for this comment. We agree and have implemented this in the manuscript. Now we use "raindrop" throughout.

29 The authors should check whether 'filter out' is more appropriate than 'filter away'.

Thank you for this comment. We agree. Now we use the word "suppress" instead of "filter away" since the rain Doppler signals are still in the spectrum after normalization.

30 Sometimes the description of the lidars is 'lidar #1/#2/#3', sometimes 'WindScanner #1/#2/#3' and sometimes 'WindScanner lidar #1/#2/#3'. To improve the reading, I suggest using the same description throughout the manuscript.

Thank you for this comment. We agree and now use lidar #1/#2/#3 both in the text and the figures.

**Bibliography**

N. Angelou, F. F. Abari, J. Mann, T. Mikkelsen, and M. Sjöholm. Challenges in noise removal from doppler spectra acquired by a continuous-wave lidar. In *Proceedings of the 26th International Laser Radar Conference, Porto Heli, Greece*, pages 25–29, 2012.

S. Davoust, A. Jehu, M. Bouillet, M. Bardon, B. Vercherin, A. Scholbrock, P. Fleming, and A. Wright. Assessment and optimization of lidar measurement availability for wind turbine control. Technical report, National Renewable Energy Lab.(NREL), Golden, CO (United States), 2014.

M. Debnath, G. V. Iungo, R. Ashton, W. A. Brewer, A. Choukulkar, R. Delgado, J. K. Lundquist, W. J. Shaw, J. M. Wilczak, and D. Wolfe. Vertical profiles of the 3-d wind velocity retrieved from multiple wind lidars performing triple range-height-indicator scans. *Atmospheric Measurement Techniques*, 10(2):431–444, 2017.

Glossary of Meteorology (June 2000). Rain. `https://glossary.ametsoc.org/wiki/Rain`, last access: 21 June 2023.

F. Guo, J. Mann, A. Peña, D. Schlipf, and P. W. Cheng. The space-time structure of turbulence for lidar-assisted wind turbine control. *Renewable Energy*, 2022.

D. Jena and S. Rajendran. A review of estimation of effective wind speed based control of wind turbines. *Renewable and Sustainable Energy Reviews*, 43:1046–1062, 2015.

L. Jin, N. Angelou, J. Mann, and G. C. Larsen. Improved wind speed estimation and rain quantification with continuous-wave wind lidar. In *Journal of Physics: Conference Series*, volume 2265, page 022093. IOP Publishing, 2022.

J. Li, X. Wang, and X. B. Yu. Use of spatio-temporal calibrated wind shear model to improve accuracy of wind resource assessment. *Applied energy*, 213:469–485, 2018.

T. Mikkelsen, M. Sjöholm, N. Angelou, and J. Mann. 3d windscanner lidar measurements of wind and turbulence around wind turbines, buildings and bridges. In *IOP Conference Series: Materials Science and Engineering*, volume 276, page 012004. IOP Publishing, 2017.

W. H. Press, W. T. Vetterling, S. A. Teukolsky, and B. P. Flannery. *Numerical recipes*. Citeseer, 1988.

S. Samadianfard, S. Hashemi, K. Kargar, M. Izadyar, A. Mostafaeipour, A. Mosavi, N. Nabipour, and S. Shamshirband. Wind speed prediction using a hybrid model of the multi-layer perceptron and whale optimization algorithm. *Energy Reports*, 6:1147–1159, 2020.

K. Träumner, J. Handwerker, A. Wieser, and J. Grenzhäuser. A synergy approach to estimate properties of raindrop size distributions using a doppler lidar and cloud radar. *Journal of Atmospheric and Oceanic Technology*, 27(6):1095–1100, 2010.

M. Türk and S. Emeis. The dependence of offshore turbulence intensity on wind speed. *Journal of Wind Engineering and Industrial Aerodynamics*, 98(8-9):466–471, 2010.

A. Van Ulden and A. Holtslag. Estimation of atmospheric boundary layer parameters for diffusion applications. *Journal of Applied Meteorology and Climatology*, 24(11):1196–1207, 1985.

T. Wei, H. Xia, J. Hu, C. Wang, M. Shangguan, L. Wang, M. Jia, and X. Dou. Simultaneous wind and rainfall detection by power spectrum analysis using a vad scanning coherent doppler lidar. *Optics express*, 27(22):31235–31245, 2019.

---

## Referee Report (RR1)

**Suppression of precipitation bias on wind velocity from continuous-wave Doppler lidars**

Liqin Jin, Jakob Mann, Nikolas Angelou, and Mikael Sjöholm

Paper ID: amt-2023-464

Iteration: Second review round.

**REVIEW COMMENTS**

**OVERALL SUMMARY**

The authors have successfully and satisfactorily addressed my review comments on a point by point basis. Therefore, I recommend publication of this manuscript. A few minor comments are given below.

**SPECIFIC COMMENTS**

Tab. 1 Caption. Consider to add: "Note geographic beam direction is relative to North and clockwise positive."

L175 Change to: "From the histogram of the maximum values of the spectra obtained during this moderate-rain minute (Fig. 8d),"

L196 Punctuation. Change to: "Then, a real atmospheric Doppler signal…"

Fig. 3 Caption. Move sentence "The dashed line in (a) indicates…" to the end of descriptor (a) as follows:

"(a) A sketch of the V52 meteorological mast with the instrumentation. Dashed lines indicates …"

Fig. 4. Please leave a space in between "… Monitor <space> (LPM)"

Fig. 7 Caption. Please rewrite for clarity as follows:

"Processing block diagram (…) 3-kHz-samples Doppler spectra. In absence of rain, the processing block diagram reduces to just the purple path including the "Divide by background spectrum" block (dashed purple line before label (2)".

(and check if this is the conveyed meaning).